# VE-cadherin in arachnoid and pia mater cells serves as a suitable landmark for in vivo imaging of CNS immune surveillance and inflammation

Josephine A. Mapunda [1,6], Javier Pareja [1,6], Mykhailo Vladymyrov [1], Elisa Bouillet [1], Pauline Hélie [1], Petr Pleskač[1], Sara Barcos[1], Johanna Andrae [2], Dietmar Vestweber [3], Donald M. McDonald [4], Christer Betsholtz[2,5], Urban Deutsch[1], Steven T. Proulx[1] & Britta Engelhardt [1] ✉

Meninges cover the surface of the brain and spinal cord and contribute to protection and immune surveillance of the central nervous system (CNS). How the meningeal layers establish CNS compartments with different accessibility to immune cells and immune mediators is, however, not well understood. Here, using 2-photon imaging in female transgenic reporter mice, we describe VE-cadherin at intercellular junctions of arachnoid and pia mater cells that form the leptomeninges and border the subarachnoid space (SAS) filled with cerebrospinal fluid (CSF). VE-cadherin expression also marked a layer of Prox1⁺ cells located within the arachnoid beneath and separate from E-cadherin⁺ arachnoid barrier cells. In vivo imaging of the spinal cord and brain in female VE-cadherin-GFP reporter mice allowed for direct observation of accessibility of CSF derived tracers and T cells into the SAS bordered by the arachnoid and pia mater during health and neuroinflammation, and detection of volume changes of the SAS during CNS pathology. Together, the findings identified VE-cadherin as an informative landmark for in vivo imaging of the leptomeninges that can be used to visualize the borders of the SAS and thus potential barrier properties of the leptomeninges in controlling access of immune mediators and immune cells into the CNS during health and neuroinflammation.

The meninges of the central nervous system (CNS) are formed by three layers, the dura mater, arachnoid mater, and pia mater, which surround and provide a protective covering of the brain and spinal cord[1,2]. The dura mater is the outermost layer and, in the brain, is directly attached to the skull. The dura mater is composed of dense fibrous tissue with a high content of collagen.

Unlike the tight blood-brain barrier (BBB), the leakiness of blood vessels in the dura mater makes the tissue more accessible to plasma components and circulating immune cells[3]. The dura mater also contains lymphatic vessels, which could serve as a drainage route for fluids, antigens, and immune cells from the CNS[4–6].

[1]Theodor Kocher Institute, University of Bern, Bern, Switzerland. [2]Department of Immunology, Genetics and Pathology, Uppsala University, Uppsala, Sweden. [3]Max-Planck- Institute for Molecular Biomedicine, Münster, Germany. [4]Cardiovascular Research Institute, UCSF Helen Diller Family Comprehensive Cancer Center, and Department of Anatomy, University of California San Francisco, San Francisco, CA, USA. [5]Department of Medicine-Huddinge, Karolinska Institute, Campus Flemingsberg, Huddinge, Sweden. [6]These authors contributed equally: Josephine A. Mapunda, Javier Pareja. ✉e-mail: britta.engelhardt@unibe.ch

The arachnoid mater is located beneath the dura mater. The cells of the arachnoid next to the dura are designated dural border cells[7,8] or "subdural neurothelium"[9] and have been referred to as part of the dura[7] or the arachnoid[10]. Arachnoid barrier cells positioned immediately beneath the dural border cells form a bona fide blood-cerebrospinal fluid barrier between the dura mater and cerebrospinal fluid (CSF) in the subarachnoid space (SAS)[7,11]. Arachnoid barrier cells are connected by claudin-11+-containing tight junctions (TJs)[7,12–14] and E-cadherin+-containing adherens junctions (AJs)[15] that limit intercellular movement of macromolecules in rodents and humans. Arachnoid barrier cells in humans and rodents also express transporters and efflux pumps comparable to endothelial cells that form the BBB and to choroid plexus epithelial cells that form the blood-CSF barrier in brain ventricles[13,16,17]. Trabeculae from the arachnoid cross the SAS toward the pia mater. These trabeculae are composed of leptomeningeal cells around a core of collagen fibers and are thought to provide a rigid scaffold for stabilizing the SAS[18].

In mice, the pia mater is formed by a single layer of flattened fibroblast-like cells covering the surface of the CNS parenchyma. The pia mater provides a barrier to erythrocytes in humans[19] and can be permeated by macromolecules such as horseradish peroxidase in rodents[20], cats and dogs[21]. The pia mater covers blood vessels in the SAS and separates the SAS from the CNS parenchyma and perivascular spaces. The pia mater is separated from the CNS parenchyma at the sites of penetrating blood vessels[1,19,22]. The subpial space filled with collagen bundles is located beneath the pia mater. The pia mater is separated from the CNS parenchyma by yet another barrier, the glia limitans, composed of the parenchymal basement membrane and astrocyte end-feet (summarized in ref. 23). The glia limitans forms a barrier to immune cells but permits passage of fluids and low molecular weight tracers from the CSF into the CNS parenchyma[24–26].

The meningeal layers separate CNS into compartments that differ in accessibility to immune cells and mediators and functional connections to the peripheral immune system. These compartments contain diverse immune cell subsets involved in CNS immune surveillance[27–30]. The dura mater harbors a rich variety of macrophages, neutrophils, B cells, and B cell precursors[27–29]. Border-associated macrophages, which differ from dura mater macrophages are found in the SAS while neuron-associated microglial cells are restricted to the CNS parenchyma[30–34].

However, recent studies have suggested that immune cells can readily cross the meningeal layers to reach dural lymphatics from the SAS[5,35] or, in reverse, travel from the dura mater towards the brain parenchyma[27,29]. Along the same lines, factors from the CSF have been suggested to influence immune cell migration from skull bone marrow niches into the dura mater[36]. Despite their novelty, all these studies have omitted consideration of the barrier properties of the arachnoid mater, which forms a barrier between the dura mater and the SAS.

Recent advances in intravital microscopy (IVM), including epifluorescence, near-infrared, and two photon (2P) imaging, have enabled in vivo visualization of some aspects of CSF circulation and immunity. However, a major limitation of fluorescence-based IVM is that "one only sees what is visible." Thus, the lack of visualization of immune cells and immune mediators in the context of the meningeal layers makes their assignment to the SAS, subpial space, or CNS parenchyma challenging and unreliable, if not impossible. This limitation prompted us to search for fluorescent reporter mice suitable for visualizing individual meningeal layers in the brain and spinal cord by 2P-IVM.

Here, we report that VE-cadherin-GFP knock-in mice, which were previously created to image vascular endothelial cells, are also useful for visualizing the arachnoid and pia mater of the brain and spinal cord. These mice used with complementary methods facilitate the exploration of the barrier properties of the leptomeninges to immune cells and mediators in health and inflammatory conditions.

## Results

### VE-cadherin identifies cellular layers of the arachnoid and pia mater

VE-cadherin-GFP knock-in mice have been widely used by us and others for both in vitro and in vivo visualization of endothelial adherens junctions (AJs)[37–39]. VE-cadherin-GFP knock-in mice express a C-terminal GFP fusion protein of VE-cadherin in the endogenous VE-cadherin locus[40]. During the course of imaging female VE-cadherin-GFP knock-in mice in our established cervical spinal cord window preparations and in skull thinning or cranial window preparations[41], we noticed an unexpected fluorescent signal on the surface of the brain and spinal cord.

Imaging the cervical spinal cords of healthy VE-cadherin GFP knock-in female mice previously injected with a vascular tracer allowed us to detect a strong GFP signal in endothelial AJs of the dorsal vein and all branching blood vessels visible on the dorsal aspect of the cervical spinal cord (Fig. 1A, B, D–G, Supplementary Movie 1). Second-harmonic generation enabled identification of the collagen type I-enriched dura mater and subpial space, as well as trabeculae bridging the subarachnoid space (SAS)[38]. To our surprise, we also observed a junctional VE-cadherin-GFP signal in two cellular layers parallel to the spinal cord surface. The first VE-cadherin-GFP+ cellular layer located immediately beneath the dura mater suggested expression of VE-cadherin in cells of the arachnoid mater, while the second VE-cadherin-GFP+ cellular layer was visible at a location corresponding to pia mater (Fig. 1A, B, D–G, Supplementary Movie 1). The VE-cadherin-GFP signal in the spinal cord arachnoid mater outlined multiple layers of VE-cadherin-GFP+ cells characterized by junctional GFP signals (Fig. 1F, Supplementary Movie 1). The single VE-cadherin-GFP+ cellular layer at the level of the pia mater was characterized by junctional localization of a bright GFP signal that outlined large cells with diameters between 30 and 70 μm (Fig. 1G, Supplementary Movie 1). The putative SAS bordered by the VE-cadherin-GFP+ layers was widest next to the dorsal vein and reduced in width towards the lateral aspects of the spinal cord (Fig. 1B, D–G, Supplementary Movie 1). Trabeculae crossing the SAS were ensheathed by VE-cadherin GFP+ cells, further supporting the conclusion that VE-cadherin is expressed by leptomeningeal cells (Fig. 1B, Supplementary Movie 1).

The developmental origins and functions of the meningeal layers of the brain and spinal cord are distinct. While the meningeal fibroblasts in the forebrain and midbrain originate from both the neural crest and mesoderm, the meninges of the hindbrain and spinal cord originate solely from mesoderm[42,43]. Considering these developmental origins, we next asked if the VE-cadherin-GFP signal would also identify leptomeningeal layers of the brain. Imaging the brain through the thinned skull of healthy VE-cadherin GFP knock-in mice revealed a strong GFP signal in AJs between endothelial cells of blood vessels at the surface of the brain (Fig. 1C, H–L). As in the spinal cord, other layers of VE-cadherin-GFP+ cells were not associated with vessels (Fig. 1C, H–L). Unlike the spinal cord, the uppermost VE-cadherin GFP+ layer over the brain was located immediately under the second-harmonic generation of the thinned skull and dura mater that was not continuous but rather consisted of individual VE-cadherin-GFP+ cells (Fig. 1J, white arrows). Beneath we identified several overlapping layers of cells with junctional VE-cadherin GFP+ signals divided into two distinguishable layers only near blood vessels, suggesting that they represent the arachnoid and pia mater. To test this assumption, we next injected VE-cadherin-GFP reporter mice with 10kDa-AF647 dextran intraarterially and 10kDa-TRITC dextran into the cisterna magna, and imaged the brain surface through the thinned skull (Fig. 1L). The intravascular tracer outlined the lumen of blood vessels in the SAS and brain parenchyma, but readily diffused throughout the dura mater, which lacks a vascular barrier, crossed the discontinuous layer of VE-cadherin-GFP+ cells, but stopped at a second VE-cadherin-GFP+ cellular layer that appeared to present a barrier for this tracer. At the same

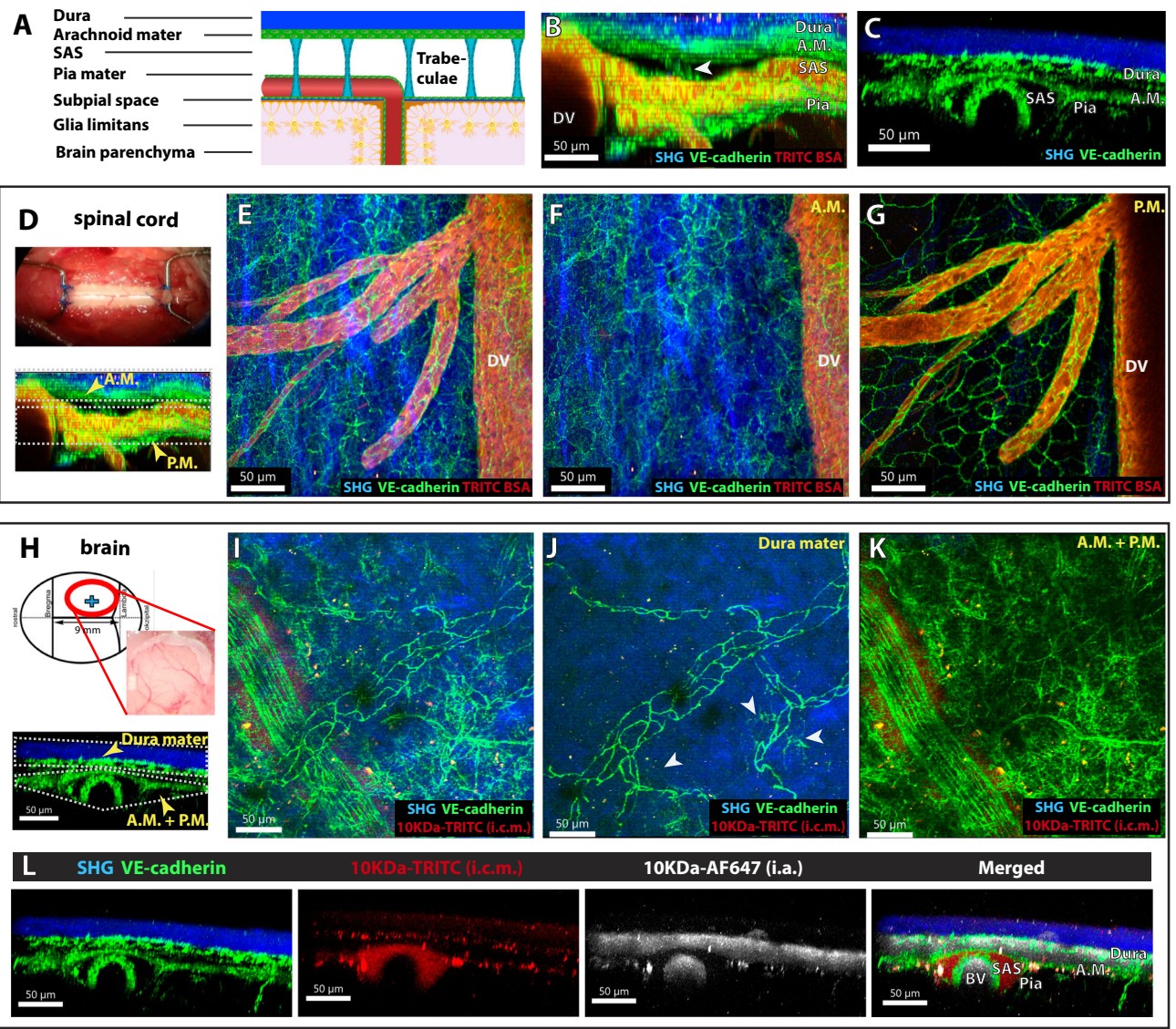

**Fig. 1 | Pia mater and arachnoid mater cells can be identified by VE-cadherin-GFP expression. A** Schematic representation of the meningeal layers on the surface of the spinal cord. **B–L** Representative images of the cervical spinal cord and brain as observed by 2P-IVM imaging of VE-cadherin GFP reporter mice via a cervical spinal cord window and skull thinning preparation, respectively. The dura mater is visible by the second-harmonic generation (blue) due to its richness in collagen type 1. VE-cadherin-GFP is shown in green. The subarachnoid space (SAS) is bordered by the VE-cadherin-GFP⁺ arachnoid and pia mater. VE-cadherin-GFP also marks blood vessels. The lumen of the spinal cord blood vessels is visible in red (**B**, **D–G**) due to systemic injection of TRITC-BSA. In the skull thinning preparation (**I–L**) the SAS is visible in red due to a cisterna magna injection of 10 kDa-TRITC dextran. Data are representative of three different mice. **B** YZ MIP of the 2P-IVM of the cervical spinal cord. Below the dura mater, a GFP signal is visible at the expected level of the arachnoid mater. The SAS is breached by VE-cadherin-GFP⁺ trabeculae (white arrowhead). A GFP signal can be seen below the SAS at the expected level of the pia mater. The lumen of the dorsal vein (DV) and subarachnoid blood vessels (red) ensheathed by VE-cadherin-GFP⁺ endothelial cells are visible. **C** YZ maximum intensity projection (MIP) of 2P-IVM of the meningeal layers on the surface of the brain of a VE-cadherin-GFP mouse. Within the dura mater, VE-cadherin-GFP⁺ blood vessels are visible. Below the dura mater, a GFP signal highlights the expected level of the arachnoid mater (AM). The VE-cadherin GFP signal can also be seen at the expected level of the pia mater. **D** Cervical spinal cord window preparation. YZ MIP of 2P-IVM of the meningeal layers on the surface of the spinal cord of a VE-cadherin-GFP mouse highlighting the arachnoid (A.M.) and pia mater (P.M.). **E** XY MIP of the meningeal layers of the spinal cord. VE-Cadherin GFP⁺ AJs on the endothelial cells are visible on the walls of the dorsal vein and branching vessels. The blood vessel

lumen is visible in red. Additional VE-cadherin GFP signal is visible outside the blood vessels. **F** XY MIP of the arachnoid mater level of the spinal cord from the region highlighted as A.M. from **D**. VE-cadherin GFP⁺ signal with junctional properties and no distinct cellular morphology that is not associated with blood vessels (red) is seen. **G** XY MIP of the pia mater from the region highlighted as P.M. from **D**. Large cells with VE-cadherin GFP⁺ AJ not associated with blood vessels (in red) are visible. **H** The skull thinning preparation and a schematic representation of the imaged anatomical location of the brain surface is shown. YZ MIP of 2P-IVM of the meningeal layers on the surface of the brain of a VE-cadherin-GFP mouse highlighting the dura, arachnoid (A.M.), and pia mater (P.M.). **I** XY MIP of the meningeal layers of the surface of the brain is shown. VE-cadherin-GFP⁺ blood vessels are visible. Additional VE-cadherin GFP signal is visible outside the blood vessels. Perivascular red tracer is visible in the SAS. **J** XY MIP of the dura mater of the brain from the region. Remnants of the skull bone and the dura mater are visualized by the second-harmonic generation (blue). The VE-cadherin GFP⁺ AJs in the blood vessels of the dura mater are visible. White arrowheads point to non-endothelial VE-cadherin-GFP⁺ cells directly below the second-harmonic generation, possibly representing individual dural border cells. **K** XY MIP of the arachnoid and pia mater of the brain from the region highlighted as A.M. + P.M. VE-cadherin-GFP⁺ AJs in the blood vessels is also clearly visible. VE-cadherin-GFP⁺ signal with no association with blood vessels is also visible. Cisterna magna infusion of a 10 kDa TRITC-Dextran allows for visualization of the SAS along a meningeal artery. **L** 2P-IVM imaging after skull thinning in a VE-cadherin-GFP mouse cisterna magna infused with a 10kDa-TRITC dextran (red) and systemically injected with a 10kDa-AF647 dextran (white). The VE-cadherin-GFP⁺ signal is shown in green. Second-harmonic generation is shown in blue. DV dorsal vein, A.M. arachnoid mater, P.M. pia mater, SAS subarachnoid space, BV blood vessel.

time, the 10 kDa-TRITC dextran tracer injected into the cisterna magna filled the space lined by VE-cadherin-GFP, underscoring that these arachnoid and pial cells border the CSF-filled SAS (Fig. 1C, L).

The combined distribution of the tracers in the context of the localization of the VE-cadherin-GFP⁺ cells on the brain surface suggests that the discontinuous layer of VE-cadherin-GFP⁺ cells directly adjacent to the dura mater identified dural border cells[7] that was breached by the 10 kDa Dextran tracer. The underlying layers included the arachnoid barrier that formed a barrier to the 10 kDa Dextran tracer and VE-cadherin-GFP⁺ cells in the arachnoid mater and pia. In the healthy brain, these leptomeningeal layers were so close to each other that the SAS could not be identified by in vivo 2P-IVM. However, they had greater separation and the SAS was visible near meningeal blood vessels (Fig. 1C, H−L). Unlike the spinal cord, in accordance to previous reports, we did not observe trabeculae crossing the SAS of the brain[44,45]. Taken together, our data show that the cells of the arachnoid mater and pia mater of the mouse brain and spinal cord express VE-cadherin and that VE-cadherin-GFP knock-in mice enable visualization of leptomeningeal barriers.

### Use of VE-cadherin-GFP knock-in mice reveals an impact of surgical preparations for brain imaging

Aiming for better visualization of the delicate junctional VE-cadherin-GFP signal in the leptomeninges, we also performed 2P-IVM of the brain using acute cranial preparations that allow imaging of the brain with the dura mater intact[46]. To our surprise, we noticed that skull thinning and acute cranial window preparations resulted in differences in the visualization of the VE-cadherin-GFP⁺ leptomeningeal layers at the brain surface (Supplementary Fig. 1). To further understand the scope of potential surgical artifacts, we imaged the brains of female VE-cadherin-GFP mice intravenously injected with a 10kDa-AF647 dextran tracer after cranial window or skull thinning preparation. Unexpectedly, we observed that, unlike the skull thinning preparations, acute cranial window preparations did not introduce a subdural space visible immediately after the surgery (Supplementary Fig. 1A). Nevertheless, we observed a widening of a subdural space over time in both acute cranial window and skull thinning preparations (Supplementary Fig. 1B−D, Supplementary Movie 2). Taken together, use of VE-cadherin-GFP knock-in mice made it possible to visualize the multiple meningeal layers and CSF spaces, as well as changes produced by surgical procedures.

### VE-cadherin is expressed by leptomeningeal cells covering the entire CNS

2P-IVM imaging through a cervical spinal cord and cranial window visualizes only small areas of the CNS. Therefore, we asked if the VE-cadherin-GFP signal could be observed in the arachnoid and pia mater throughout the CNS. To this end, we performed *post mortem* epifluorescence imaging of the brain and spinal cord of healthy VE-cadherin GFP knock-in mice and wild-type C57BL/6J mice.

In VE-cadherin-GFP knock-in mice, the GFP signal was visible over the entire spinal cord and brain surface (Fig. 2). Imaging at higher magnification (×100) revealed the expected junctional localization of VE-cadherin-GFP at endothelial AJs (Fig. 2). In addition, we observed the GFP signal over the dorsal (Fig. 2A−C) and ventral (Fig. 2D−F) surfaces of the cervical (Fig. 2A, D), thoracic (Fig. 2B, E), and lumbar (Fig. 2C, F) regions of the spinal cord of VE-cadherin-GFP knock-in mice. No GFP signal was found in wild-type C57BL/6J control mice (Supplementary Fig. 2A−F). In regions of the spinal cord where the arachnoid mater was disrupted during the tissue removal procedure, we observed a bright GFP signal at junctions that outlined large cells in the remnant of pia mater (Fig. 2C). Similarly, we observed VE-cadherin GFP signal covering the dorsal surface of the brain (Fig. 2G), specifically the olfactory bulbs (Fig. 2H), brain cortex (Fig. 2I), and cerebellum (Fig. 2J), and covering the ventral surface of the brain (Fig. 2K).

Taken together, imaging of VE-cadherin-GFP knock-in and wild-type C57BL/6J female mice confirmed the specificity of the VE-cadherin-GFP signal in the leptomeningeal layers covering the entire brain and spinal cord and provided further evidence of VE-cadherin expression in cells of the arachnoid and pia mater.

### VE-cadherin is not restricted to endothelial cell junctions

VE-cadherin has been described as an endothelial cell-specific cadherin essential for the formation and maintenance of endothelial AJs[47]. To obtain further evidence that the VE-cadherin-GFP signal in the leptomeninges was not from vascular endothelial cells, we stained sections of brain and spinal cord from female VE-cadherin-GFP knock-in mice for PECAM-1, a cell adhesion and signaling molecule enriched at endothelial cell junctions[48].

Confocal imaging of 100-μm-thick sections documented that PECAM-1 immunoreactivity accompanied by VE-cadherin-GFP fluorescence was restricted to blood vessels in the spinal cord (Fig. 3A) and brain (Fig. 3B) of healthy VE-cadherin-GFP knock-in mice. In the same sections, VE-cadherinGFP fluorescence was not accompanied by PECAM-1 staining in leptomeningeal cells of the spinal cord (Fig. 3A) or brain (Fig. 3B).

We next sought to determine whether the VE-cadherin-GFP signal we observed outside endothelial AJs reflected endogenous VE-cadherin expression rather than an artifact of the transgenic mice. We stained brain and spinal cord sections from wild-type C57BL/6J mice for VE-cadherin and PECAM-1 immunoreactivities. Confocal imaging of 20 μm brain and spinal cord sections from these mice further documented PECAM-1 and VE-cadherin immunostaining in blood vessels, but only VE-cadherin staining was found in the leptomeninges of the spinal cord (Fig. 3C) and brain (Fig. 3D). These findings confirm the presence of VE-cadherin staining in leptomeningeal cells that is separate from VE-cadherin in endothelial cells.

### Arachnoid and pia mater cells of the brain and spinal cord stain positive for VE-cadherin

To further verify if the observed meningeal VE-cadherinGFP signal associates with cells of the arachnoid and pia mater, we next performed immunofluorescence stainings for molecules reported to be expressed by arachnoid and pial mater fibroblast-like cells.

ER-TR7 is located in the cytoplasm of reticular fibroblasts[49] and is an extracellular matrix component widely used as a marker for fibroblasts[50] and to detect reticular fibroblast-like cells in the meninges[51]. To determine whether the VE-cadherin-GFP signal in the arachnoid and pia mater colocalizes with ER-TR7, we stained brain and spinal cord sections of VE-cadherin-GFP reporter mice for ER-TR7 immunoreactivity. Confocal imaging showed ER-TR7 immunoreactivity overlapped the VE-cadherin-GFP signal in leptomeningeal layers of brain and spinal cord (Fig. 4A). ER-TR7 staining also colocalized with VE-cadherin-GFP⁺ cells in meningeal blood vessels entering the CNS parenchyma (Fig. 4A), consistent with the the concept that the pia mater ensheathes blood vessels branching into the CNS parenchyma[1,19,22].

Activated leukocyte cell adhesion molecule (ALCAM) is another marker reported to be expressed by leptomeningeal cells[52]. Immunostaining showed prominent ALCAM staining in VE-cadherin-GFP⁺ layers of leptomeninges in sections of brain and spinal cord in VE-cadherin-GFP reporter mice (Fig. 4B). No ALCAM immunostaining was detected in VE-cadherin-GFP⁺ blood vessels of the brain, and only scattered ALCAM⁺ cells accompanied spinal cord vessels (Fig. 4B).

E-cadherin is a component of AJs that interconnect arachnoid barrier cells[53,54]. Hence, we asked whether E-cadherin and VE-cadherin are co-expressed in arachnoid barrier cells. To this end, we performed immunofluorescence stainings of the spinal cord and brain sections of VE-cadherin-GFP reporter mice for E-cadherin. As expected, we observed prominent immunostaining for E-cadherin identifying

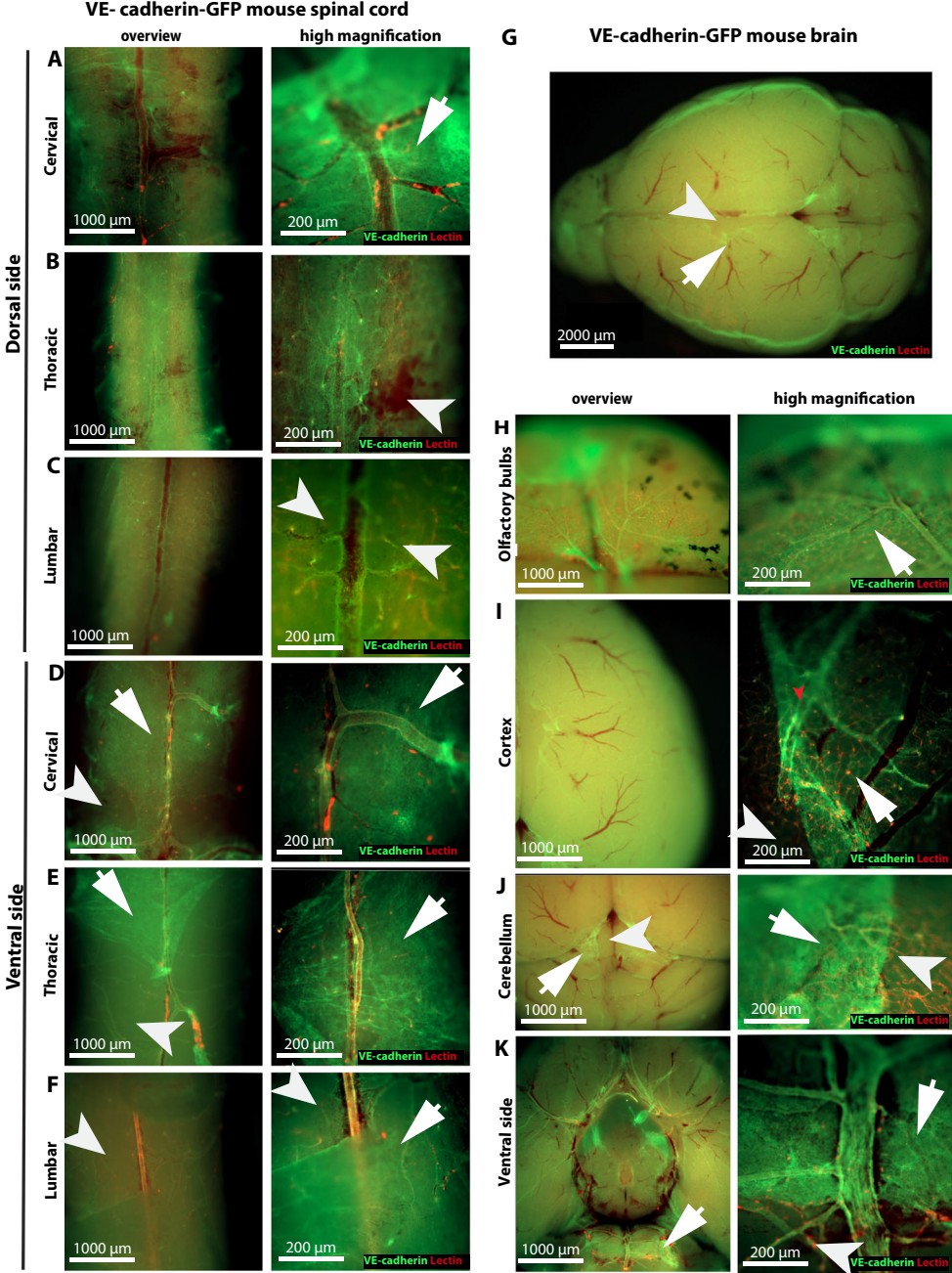

**Fig. 2 | VE-cadherin-GFP expression is detected in meningeal layers covering the entire CNS.** Representative images of ex vivo imaging of the leptomeningeal layers covering the different regions of the brain and spinal cord of a VE-cadherin GFP knock-in reporter mouse taken with an epifluorescence microscope. VE-cadherin GFP is visible in green. Tomato lectin was injected intravenously to visualize the blood vessels (in red). The region covered by the meninges is highlighted by a white arrow. A white arrowhead highlights regions where the meninges are ripped off. Data are representative of three different mice. **A**–**F** Overviews (left) and high magnification (right) representative images of VE-cadherin GFP⁺ arachnoid barrier cells (Fig. 4C). E-cadherin appeared to overlap some meningeal layers covering the **A** dorsal and **D** ventral cervical, **B** dorsal and **E** ventral thoracic, and **C** dorsal and **F** ventral lumbar regions of the spinal cord that is not associated with blood vessels (in red). **G** Representative image of an overview of the VE-cadherin GFP⁺ meningeal layer covering the whole brain surface dorsally. Blood vessels are visible in red. **H**–**K** Overviews (left) and high magnification (right) representative images of VE-cadherin GFP⁺ meningeal layer covering the dorsal side of **H** the olfactory bulbs **I** the cortex **J** the cerebellum, and **K** ventral side of the brain, which is not associated with blood vessels (red).

arachnoid barrier cells (Fig. 4C). E-cadherin appeared to overlap some of the more widespread VE-cadherin-GFP signal in the arachnoid mater in brain and spinal cord of VE-cadherin-GFP reporter mice (Fig. 4C). However, examination of individual optical sections, which limited superimposition of staining in different optical planes, revealed that E-cadherin and VE-cadherin-GFP were largely separate spatially in the arachnoid, where E-cadherin was restricted to barrier cells of the outer arachnoid, and VE-cadherin-GFP was largely limited to cells of the inner arachnoid (Fig. 4C). Importantly, no E-cadherin staining was found in VE-cadherin GFP⁺ cells of the pia mater (Fig. 4C).

The distribution of VE-cadherin in the arachnoid mater was further characterized by staining thin coronal sections of the brain and spinal cord of VE-cadherin-GFP knock-in mice for the tight junction protein claudin-11, which is known to be expressed by arachnoid barrier cells[14]. Claudin-11 staining was found in E-cadherin⁺ barrier cells of the outer arachnoid that had little or

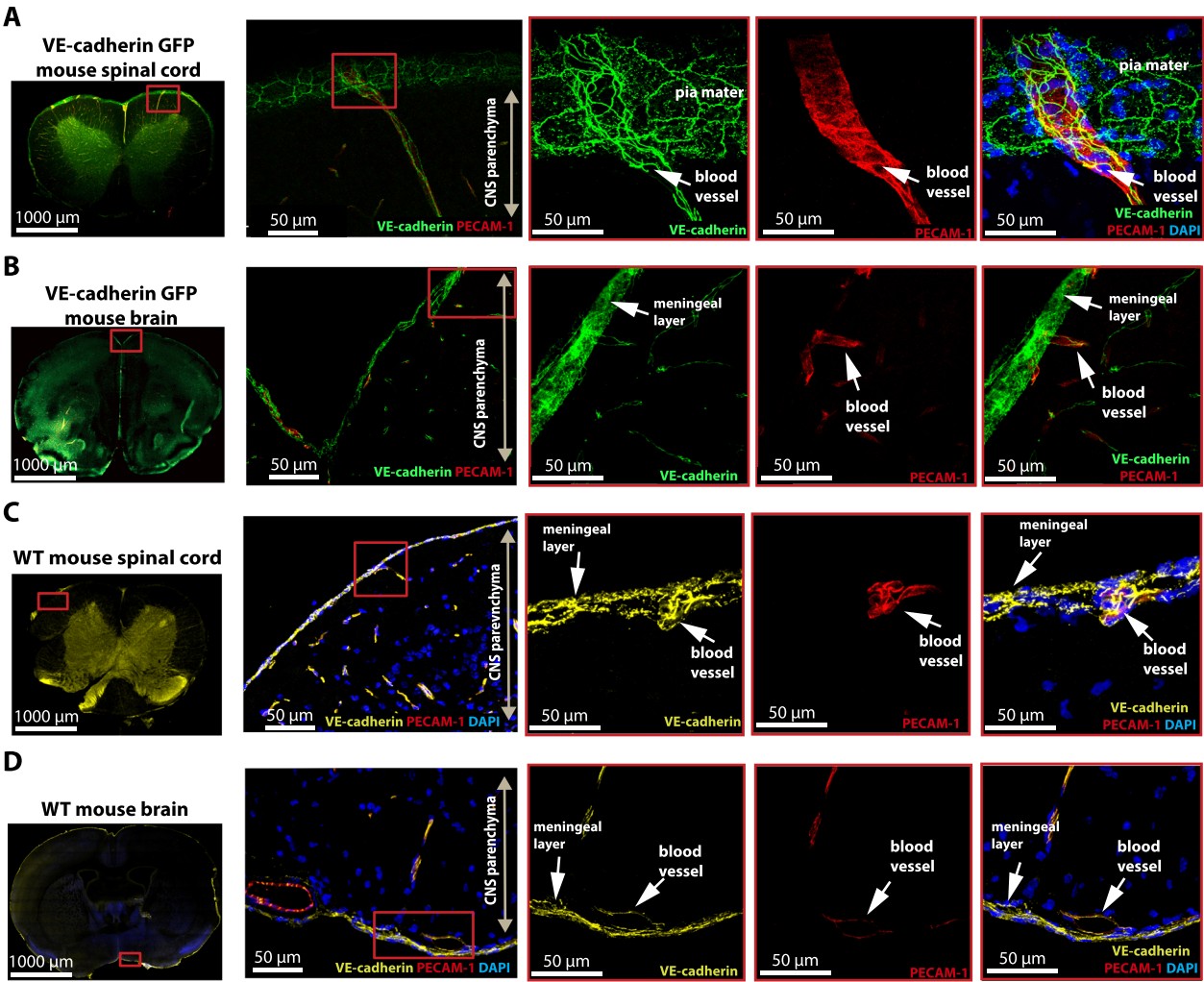

**Fig. 3 | VE-cadherin is not restricted to endothelial cell junctions.** Confocal imaging of 100 µm thick brain and spinal cord sections of a healthy VE-cadherin GFP knock-in reporter mouse and 20 µm thick brain and spinal cord sections of wild-type C57BL/6 J. PECAM-1 (red) and/or VE-cadherin (yellow) immunostaining was performed. DAPI (blue) stains the nucleus. Magnification ×40. **A–D** XY MIP of the overview (left) and zoomed-in (right) representative images of the meningeal layers of the **A–C** spinal cord and **B–D** brain. **A, B** Endogenous VE-cadherin GFP+ signal in the endothelial AJs of the meningeal and parenchymal blood vessels is seen in green. Additional VE-cadherin GFP signal is visible outside the blood vessel walls (red) on the surface of the brain and spinal cord. **C, D** VE-cadherin staining in the endothelial AJs of the meningeal and parenchymal blood vessels is seen in yellow. Additional VE-cadherin staining (yellow) is visible outside the blood vessel walls (red) on the surface of the brain and spinal cord. Images are representative of a total of three mice imaged in a total of three experiments.

no VE-cadherin-GFP signal in the brain and spinal cord (Supplementary Figs. 3, 4). Claudin-11 was also found in spinal cord white matter, where it is known to be expressed in oligodendrocytes and in tight junctions of myelin sheaths[55,56] (Supplementary Fig. 3).

We interpret these findings as showing strong VE-cadherin expression in cells of the pia and inner arachnoid but little or none in arachnoid barrier cells, which instead express E-cadherin and claudin-11 in the mouse brain and spinal cord (Supplementary Fig. 4A, arrowheads). Taken together, these data highlight the presence and distinctive distribution of VE-cadherin in the pia and inner arachnoid of the CNS.

## VE-cadherin is located at adherens junctions in arachnoid and pia mater cells

To determine whether VE-cadherin in leptomeningeal cells has the features of adherens junctions elsewhere, we stained brain sections from female VE-cadherin-GFP knock-in mice for α- and β-catenin, which are intracellular components of AJs in meningeal fibroblasts, based on mRNA expression analysis[10]. As expected, the VE-cadherin-

GFP signal was strongly associated with α-catenin and β-catenin staining at AJs in vascular endothelial cells (Fig. 5A, B). Staining for α-catenin and β-catenin was also associated with VE-cadherin-GFP in leptomeningeal cells (Fig. 5A, B). Further analysis revealed a punctate pattern of VE-cadherin-GFP in many leptomeningeal cells (Fig. 5C, white arrowheads, Supplementary Movie 3), which differed from the linear pattern in endothelial cells (Fig. 5C, yellow arrowheads, Supplementary Movie 3). In some leptomeningeal cells, VE-cadherin-GFP had a linear distribution or a mixture of punctate and linear patterns, as observed in VE-cadherin-GFP reporter mice examined by immunostaining and by 2P-IVM in skull-thinning preparations (Supplementary Movie 3).

To bridge the findings by immunostaining of junctional proteins in the leptomeninges to the ultrastructural features of tight junctions, adherens junctions, and gap junctions, as originally described[7,20,57,58], we examined the meninges of normal adult mice by transmission electron microscopy (TEM). Consistent with previous reports[7], cells of all layers of arachnoid were interconnected by AJs, identified as regions of apposed plasma membranes that had cytoplasmic densities and were separated by a narrow intercellular

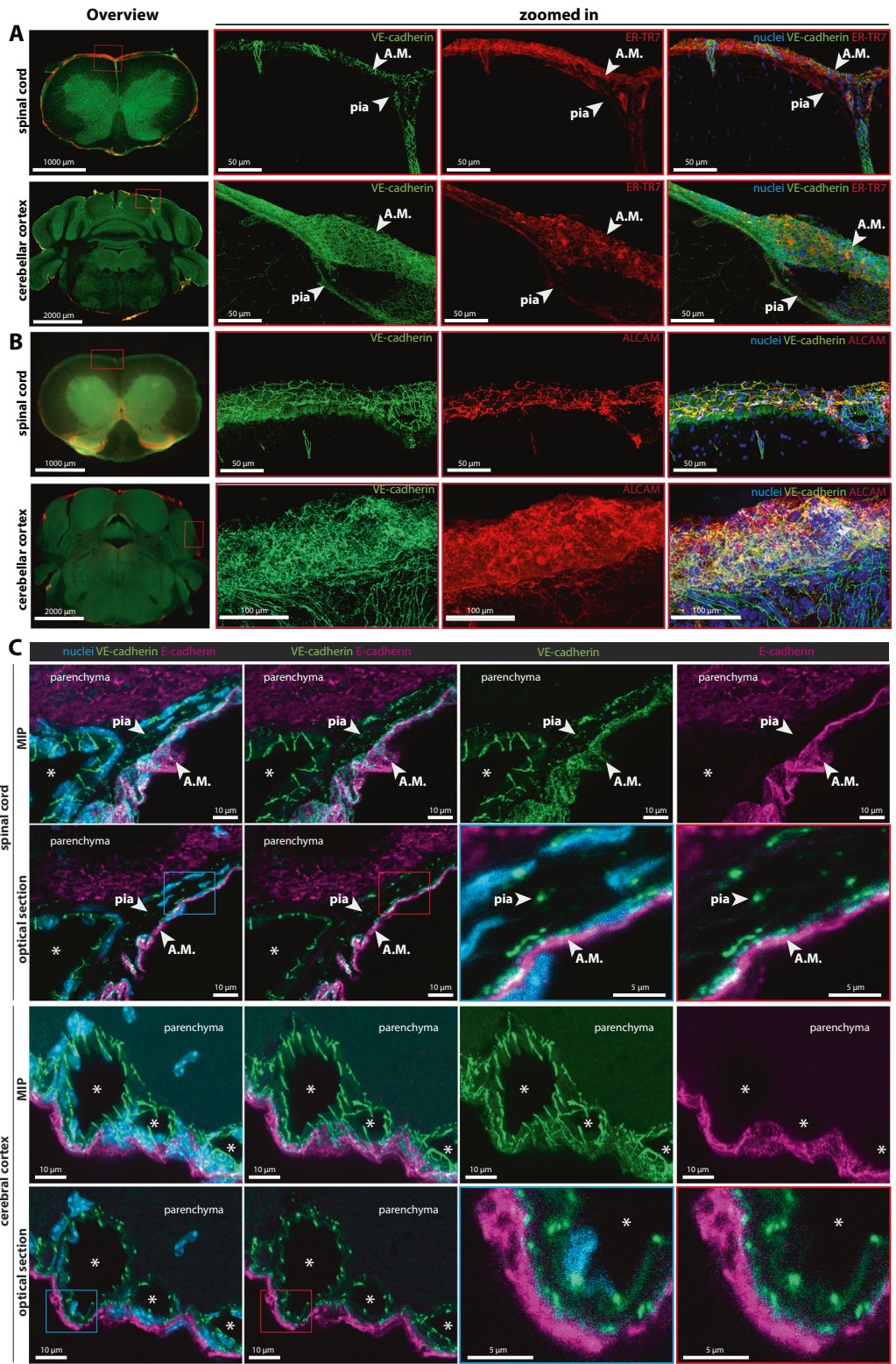

space (Fig. 5D). Unlike the broad distribution of AJs throughout the arachnoid, only the outer barrier layers of arachnoid cells were connected by TJs, where focal regions of adjacent plasma membrane outer leaflets appeared fused (Fig. 5D), as described previously[7]. In addition, some arachnoid cells were joined by gap junctions (Fig. 5D). Although the TJs observed fit the conventional TEM criteria for these junctions, TEM tracer studies and freeze fracture replicas would be needed for confirmation[7]. Another distinctive ultrastructural feature of the arachnoid was the presence of

basement membrane on the surface of cells next to wide extracellular spaces (Fig. 5D).

Together, the observations provide evidence of the contribution of VE-cadherin to AJs that interconnect cells of the pia and inner arachnoid. Consistent with the observations at the ultrastructural level the continuous VE-cadherin-GFP pattern seems to identify AJs within a single layer of endothelial or leptomeningeal cells, while the spotty pattern observed for VE-cadherin-GFP in the arachnoid mater may rather result from AJs between multiple layers of cells.

**Fig. 4 | VE-cadherin expression in meningeal cells of the brain and spinal cord.** Confocal microscopic images of 100-μm (**A**, **B**) and 20-μm (**C**) sections of the spinal cord and brain of healthy VE-cadherin-GFP knock-in reporter mice after immunostaining for known markers for the arachnoid (E-cadherin, ALCAM, ER-TR7) and pia mater (ER-TR7 and ALCAM). DAPI (blue) stains the nucleus. Magnification ×40 (**A**, **B**) and ×63 (**C**). **A**, **B** XY MIP of the overview (left) and zoomed-in (right) stained spinal cord and brain sections. In the overview images, staining for ER-TR7 and ALCAM can be seen in meningeal layers over the surface of the brain and spinal cord. **A** In the zoomed-in images (right), the VE-cadherinGFP signal overlaps ER-TR7 staining (red) in the arachnoid mater (A.M.) and pia mater. A layer of ER-TR7-positive cells (red) also covers pial blood vessels (green) entering the CNS parenchyma. Data are representative of three independent experiments. **B** VE-cadherin GFP signal in the arachnoid and pia mater overlaps ALCAM staining (red) over the brain and the spinal cord. Scale bars of the whole spinal cord and brain section overviews = 1000 μm and 2000 μm, respectively, zoomed-in (right) = 50 and 100 μm as indicated. Data are representative of three independent experiments. **C** VE-cadherin-GFP (green) signal of the arachnoid mater is juxtaposed to E-cadherin (magenta) immunoreactivity of decalcified vertebral column and head. No E-cadherin immunoreactivity is visible in the pia mater. White asterisks mark vessel lumens. Optical section = 0.16 μm. MIP maximal intensity projection. Scale bars = 5 and 10 μm as indicated. Data are representative of six independent experiments. A.M. = arachnoid mater.

## The arachnoid mater includes a layer of VE-cadherin⁺/Prox1⁺ cells

As a recent study described division of the SAS into an upper and a lower compartment by a Prox1-expressing meningeal cell layer[59] we crossed our VE-cadherin-GFP knock-in mice with Prox1-tdTomato reporter mice to analyze the localization of the respective meningeal layers by immunostaining and 2P-IVM imaging. Confocal microscopic imaging revealed that the Prox1-tdTomato⁺ layer coincided with but was less extensive than VE-cadherin-GFP⁺ layers of arachnoid—and was absent in the pia mater—in sections of spinal cord and brain of VE-cadherin-GFP; Prox1-tdTomato double reporter mice (Fig. 6A, B). The Prox1-tdTomato⁺ cells within VE-cadherin-GFP⁺ layers of the inner arachnoid were distinct from and located directly beneath E-cadherin⁺ arachnoid barrier cells (Fig. 6A, B). The findings are evidence that Prox1⁺ cells are a subset of VE-cadherin⁺ cells of the arachnoid located beneath the arachnoid barrier cells.

2P-IVM imaging confirmed that Prox1⁺ cells were located within VE-cadherin-GFP⁺ layers of the arachnoid in the spinal cord and brain of VE-cadherin-GFP; Prox1-tdTomato mice (Fig. 6C–F, Supplementary Movie 4). YZ projections of 2P-IVM images of spinal cord and brain documented the presence of Prox1-tdTomato⁺ cells among VE-cadherin-GFP⁺ arachnoid cells (Fig. 6C, Supplementary Movie 4). Additional Prox1-tdTomato⁺ cells were located below VE-cadherin-GFP⁺ cells of the pia mater of the spinal cord, as expected from Prox1 expression in some precursors of oligodendrocytes and neurons[60,61] in the CNS (Fig. 6C, D, Supplementary Movie 4). XY projections of 2P-IVM images showed that Prox1-tdTomato⁺ cells formed a discontinuous layer among VE-cadherin-GFP⁺ cells over the surface of the spinal cord, providing further evidence of localization within the arachnoid mater (Fig. 6C, D, Supplementary Movie 4). In contrast, Prox1-tdTomato⁺ cells formed a more continuous layer within the VE-cadherin-GFP⁺ arachnoid cells over the brain (Fig. 6C, E).

To probe the barrier function of the Prox1-tdTomato⁺ cells in the arachnoid mater we next imaged the brain of VE-cadherin-GFP; Prox1-tdTomato mice systemically injected with 10kDa-AF647 dextran tracer. In addition to labeling the lumen of the BBB forming blood vessels, the tracer readily extravasated from the blood vessels of the dura mater (Fig. 6F). We did not observe any tracer breaching the outer VE-cadherin-GFP⁺ meningeal layer harboring the Prox1-tdTomato⁺ cells, confirming their association with the arachnoid barrier cells (Fig. 6F).

Taken together, these findings provide evidence that Prox1-tdTomato⁺ cells are located within the VE-cadherin-GFP⁺ layers of inner arachnoid, separate from and beneath the arachnoid barrier. The data are inconsistent with the report that Prox1⁺ cells divide and form a barrier between the two compartments of the SAS[59]. Instead, Prox1⁺ cells are an integral part of the inner arachnoid that lacks a barrier function.

## The VE-cadherin-GFP reporter mouse allows for visualizing CNS zoning in vivo

The brain barriers divide the CNS in compartments with different accessibility for immune mediators and immune cells[62]. Having identified VE-cadherin as a landmark for the arachnoid and the pia mater, we used female VE-cadherin-GFP reporter mice to assess the barrier properties of the leptomeninges that contribute to this CNS zonation. We first asked if the arachnoid and pia mater create barriers for macromolecules introduced into the CSF. To this end, we injected 40 kDa TRITC-Dextran or TRITC-BSA (ca. 66 kDa) into the cisterna magna while imaging the cervical spinal cord by 2P-IVM (Fig. 7A, Supplementary Movie 5). The fluorescent tracers were detected in the SAS of the spinal cord between 10 to 20 min after cisterna magna infusion (Fig. 7A, Supplementary Movie 5).

To determine if the arachnoid mater and pia mater block movement of the tracers outside the SAS, we compared the mean fluorescence intensity of the tracers over time in different compartments of the CNS, namely the dura mater, SAS, subpial compartment, and spinal cord parenchyma. To this end, we used the second-harmonic generation signal from the collagen type I fibers in the dura mater and directly under the pia mater, as landmarks to segment both the dura mater and the subpial compartment. Tracer movement into the CNS parenchyma was determined from the fluorescent signal at a distance of 20-50 μm below the VE-cadherin-GFP⁺ pia mater (Fig. 7B). Full dynamic quantification of cisterna magna injected tracer over time revealed that the arachnoid mater forms a true barrier for 40 kDa Dextran and BSA (ca. 66 kDa), as no fluorescent signal increase was detected in the dura mater (Fig. 7C). At the same time, a prominent increase of fluorescent signal was detected in the subpial compartment but not in the CNS parenchyma, suggesting that the glia limitans - not the pia mater - served as a size-selective barrier to these CSF-derived tracers (Fig. 7C). Interestingly, after injection of TRITC-BSA, we also observed the appearance of patchy, cell-sized fluorescent signals in the SAS (Fig. 7A), consistent with active uptake of the tracer by border-associated macrophages and their contribution to CNS zonation.

We next used 2P-IVM imaging to determine whether the arachnoid mater also creates a barrier to small molecular weight tracers. At 15 to 20 min after infusion into the cisterna magna, a 3 kDa TRITC-Dextran was detected in the SAS of the spinal cord, with the same arrival kinetics as for the macromolecular tracers (Fig. 7A, C). Dynamic measurements showed that the 3 kDa TRITC dextran crossed the pia mater and diffused into the spinal cord parenchyma, but little to no fluorescent signal reached the dura mater, consistent with the barrier properties of the arachnoid mater. A small, delayed increase in the fluorescent signal was observed in the dura mater (Fig. 7C). This delayed appearance could be due to the tracer entering the dura from the bloodstream, as CSF tracers reach the peripheral blood circulation already 20 to 25 min after infusion into the cisterna magna[63].

Finally, we sought to assess leptomeningeal barrier properties to physiological molecules such as chemokines. After learning that 15μg CCL19 conjugated to Dy649P1 injected into the cisterna magna failed to produce any visible fluorescent signal in the SAS of the spinal cord, we performed a new set of experiments to follow the distribution of 10 kDa TRITC-Dextran, comparable in size to

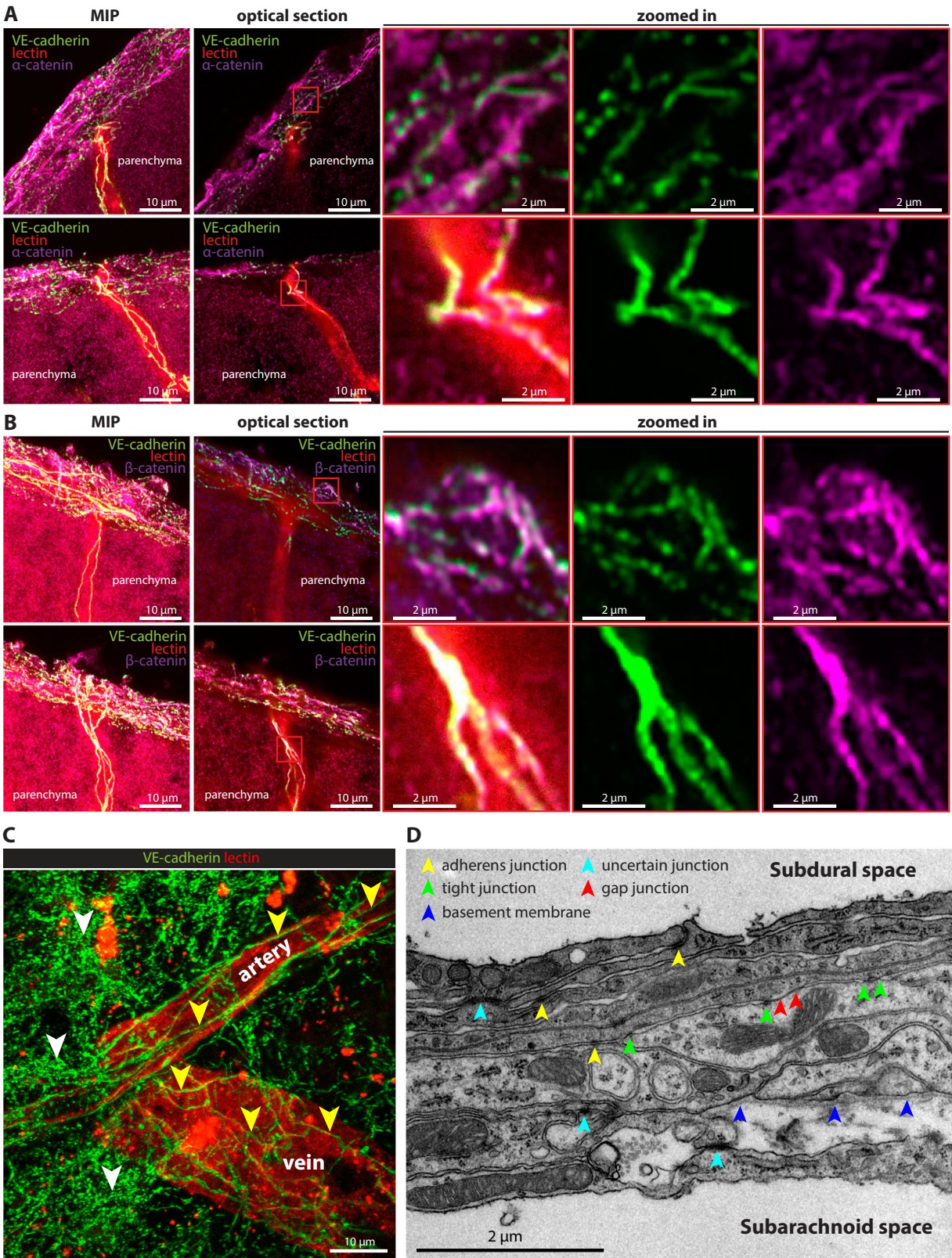

chemokines, after cisterna magna infusion (Fig. 7A). Despite differences observed with respect to the dynamics of tracer arrival and signal intensity (Fig. 7D), we consistently saw that 10 kDa TRITC-Dextran entered the subpial compartment but not the dura mater or spinal cord parenchyma (Fig. 7D, E). These findings confirm the barrier properties of the arachnoid mater in healthy mice and demonstrate the reproducibility of our experimental approach.

## Imaging alterations in CNS zoning during neuroinflammation in vivo

We next determined whether the leptomeningeal barrier was impaired in neuroinflammation. To this end, we induced active experimental autoimmune encephalomyelitis (aEAE) in female VE-cadherin-GFP reporter mice. In aEAE, CD4[+] encephalitogenic T cells enter the CNS and cause BBB breakdown, immune cell infiltration, and clinical

**Fig. 5 | VE-cadherin at adherens junctions between cells of the arachnoid and pia mater. A, B** Representative images of 20-µm coronal frozen sections of the brain of a healthy VE-cadherin-GFP reporter mouse perfused with tomato lectin-DyLight 594 (red), acquired with a confocal laser scanning microscope equipped with the Airyscan detector. Brains were cut -1 mm caudal to the bregma. Immunostaining for α-catenin (**A**) and β-catenin (**B**) is magenta. Colocalization of the VE-cadherin-GFP signal (green) with α-catenin or β-catenin immunoreactivity (magenta) in the leptomeninges is white. Optical section = 0.16 µm. MIP, maximal intensity projection. Scale bars = 10 µm (left 2 columns); 2 µm (right 3 columns of zoomed-in images). Images are representative of three independent experiments. **C** Representative image of a 50-µm transverse frozen section of the decalcified head of a healthy VE-cadherin-GFP (green) reporter mouse perfused with tomato lectin-DyLight 594 (red), acquired with a confocal laser scanning microscope. Continuous linear patterns of VE-cadherin-GFP$^+$ junctions in vascular endothelial cells (yellow arrowheads) differ from the more punctate pattern of VE-cadherin-GFP$^+$ junctions between leptomeningeal cells (white arrowheads). Longitudinally oriented arterial endothelial cells, aligned with the direction of blood flow, are readily distinguished from polygonal endothelial cells in a venule. The XY MIP image was compiled from 20 optical sections, each 0.17 µm thick. Scale bar = 10 µm. Images are representative of three independent experiments. **D** TEM image of the cellular layers of the arachnoid with intercellular junctions marked by colored arrowheads. Tight junctions (*zonula occludens*), adherens junctions (intermediate junctions or *zonula adhaerens*), and gap junctions were identified by criteria described in the Methods. Blue arrowheads mark the basement membrane of a region of inner arachnoid cells facing a wide intercellular space. Subarachnoid and subdural spaces are labeled accordingly. Scale bar = 2 µm. Image is representative of TEM images of arachnoid from three mice.

disease[25,64,65]. Either 3-, 10- or 40-kDa TRITC-Dextran or TRITC-BSA was infused into the cisterna magna during 2P-IVM of the cervical spinal cord at clinical onset of aEAE (day 14-15 post-immunization (p.i.), clinical score +). During aEAE, 10- and 40-kDa TRITC Dextrans and TRITC-BSA were readily detected in the SAS of the spinal cord at 15 min after infusion into the cisterna magna, as observed in healthy mice (Fig. 7B). In contrast, 3 kDa TRITC-Dextran was not detected in the SAS of the spinal cord but rather appeared in the lumen of subpial veins and dorsal vein at 10 min after infusion, suggesting that the tracer crossed impaired CSF barriers and entered the venous bloodstream of the spinal cord (Fig. 8A). At 45 min after infusion, 3 kDa TRITC-Dextran was observed in the dura mater, probably entering via the systemic bloodstream (Supplementary Movie 6).

Interestingly, none of the larger tracers injected (10 kDa dextran, 40 kDa dextran and BSA) could be detected in the dura mater in mice at the onset of aEAE indicating that at this timepoint arachnoid barrier properties are still intact. As already observed in healthy mice, all tracers readily crossed the pia mater reaching the subpial space but not the parenchyma (Fig. 8B).

Taken together, these observations provide evidence that the leptomeningeal layers form size-selective barriers for CSF-derived molecules, where the arachnoid mater contained a barrier to low molecular weight tracers, but the pia mater did not. In neuroinflammation, the arachnoid mater retained its barrier properties for low molecular weight tracers, but the pia mater permitted passage of CSF-derived tracers independent of their size. Together with the finding of the rapid appearance of 3kDa-Dextran in the spinal cord veins, our observations underscore that the barrier properties of the leptomeninges determine the distribution of CSF-derived molecules that enter the CNS parenchyma and reach the periphery.

**VE-cadherin-GFP knock-in mice enable observation of changes in leptomeningeal CSF spaces in health and neuroinflammation**
Despite the slow rate (1 µl/min) of infusion into the cisterna magna at a volume of only 2.5 µl, we observed a separation of the VE-cadherin-GFP$^+$ arachnoid and pial layers of the spinal cord prior to the arrival of the fluorescent tracers, suggesting that even this small infusion volume can widen the SAS (Fig. 9A–C, Supplementary Movie 5). To quantify the widening of the spinal cord SAS after tracer infusion into the cisterna magna, we measured the cross-sectional area of the SAS at the cervical spinal cord from 2P-IVM images acquired prior to and 20 min after cisterna magna infusion. The area of the SAS was defined as the space outlined by the VE-cadherin-GFP signal from the arachnoid mater, pia mater, and wall of the dorsal vein (Fig. 9A). Typically, the width of the cervical spinal cord SAS was largest near the dorsal vein and became narrower towards the lateral aspect of the spinal cord. We observed that infusion of tracer into the cisterna magna increased cervical spinal cord SAS area 2.05-fold in healthy mice and 1.35-fold in mice at the onset of aEAE (Fig. 9B, C). At the same time, we noted a 3.46-fold enlarged area of the cervical spinal cord SAS in mice

at the onset of aEAE when compared to healthy VE-cadherin-GFP reporter mice already prior to cisterna magna infusion of the tracer (Fig. 9D, E). Importantly, the subpial space bordered by VE-cadherin-GFP$^+$ pia mater cells and the dorsal vein was widened at the onset of clinical aEAE, but was less prominent at the chronic phase of aEAE (Fig. 9D, E, Supplementary Movies 7, 8). These observations indicate that CSF-filled compartments of the spinal cord undergo significant changes in volume in aEAE. These volume changes could impact CSF flow, the distribution of immune mediators, and trafficking of immune cells in the CNS.

**VE-cadherin-GFP reporter mice enable visualization of immune cell trafficking across the pia mater in vivo**
Peripherally activated T cells can cross the BBB or choroid plexus to reach the CSF-filled SAS or perivascular spaces during CNS immune surveillance (summarized in[66]). Therefore, we next determined whether the leptomeninges serve as a barrier to peripherally activated T cells that do not recognize CNS antigens but cross the BBB or choroid plexus epithelium during CNS immune surveillance or neuroinflammation. To address this issue, we used ODC-OVA mice, which express ovalbumin (OVA) as a neo-self-antigen in oligodendrocytes that is visible to CD8$^+$ T cells or B cells[67]. CNS autoimmune inflammation in ODC-OVA mice is induced by the adoptive transfer of naive TCR tg ovalbumin-specific CD8$^+$ T cells (OT I) cells followed 24 hr later by peripheral infection by OVA expressing lymphocytic choriomeningitis virus (LCMV-OVA)[68]. We crossed ODC-OVA mice with VE-cadherin-GFP knock-in mice and naive OT I cells expressing tdTomato were injected i.v. into either ODC-OVA; VE-cadherin-GFP knock-in mice or VE-cadherin-GFP knock-in reporter mice. 7 days after LCMV-OVA infection, when ODC-OVA mice manifested clinical symptoms, we performed 2P-IVM of the cervical spinal cord. We observed circulating CD8$^+$ T cells and CD8$^+$ T cells in the SAS bordered by VE-cadherin-GFP$^+$ cells of the arachnoid and pia mater (Fig. 10, Supplementary Movies 9, 10). While few tdTomato$^+$ CD8$^+$ T cells were associated with the VE-cadherin-GFP$^+$ arachnoid cells or meningeal cells covering trabeculae, significantly more CD8$^+$ T cells interacted with VE-cadherin-GFP$^+$ expressing cells in the pia mater (Fig. 10A–C, Supplementary Movies 9, 10). Here, the GFP signal from cellular junctions enabled us to localize most CD8$^+$ T cells to either above or below the pia mater (Fig. 10D, E). However, due to the limited axial resolution of 2P-IVM, occasional events of flattened CD8$^+$ T cells, could not be assigned with certainty to be localized above or below the pia mater. Flattening of CD8$^+$ T cells was observed preceding transmigration, which suggests that some CD8$^+$ T cells underwent transcellular and paracellular diapedesis across the VE-cadherin-GFP$^+$ pial cell layer.

The observed interactions between CD8$^+$ T cells and cells of the pia mater suggest that CD8$^+$ T cells recognize molecular cues on the pia mater, allowing them to find sites permissive for diapedesis (Fig. 10F, Supplementary Movie 10). In contrast, we did not observe

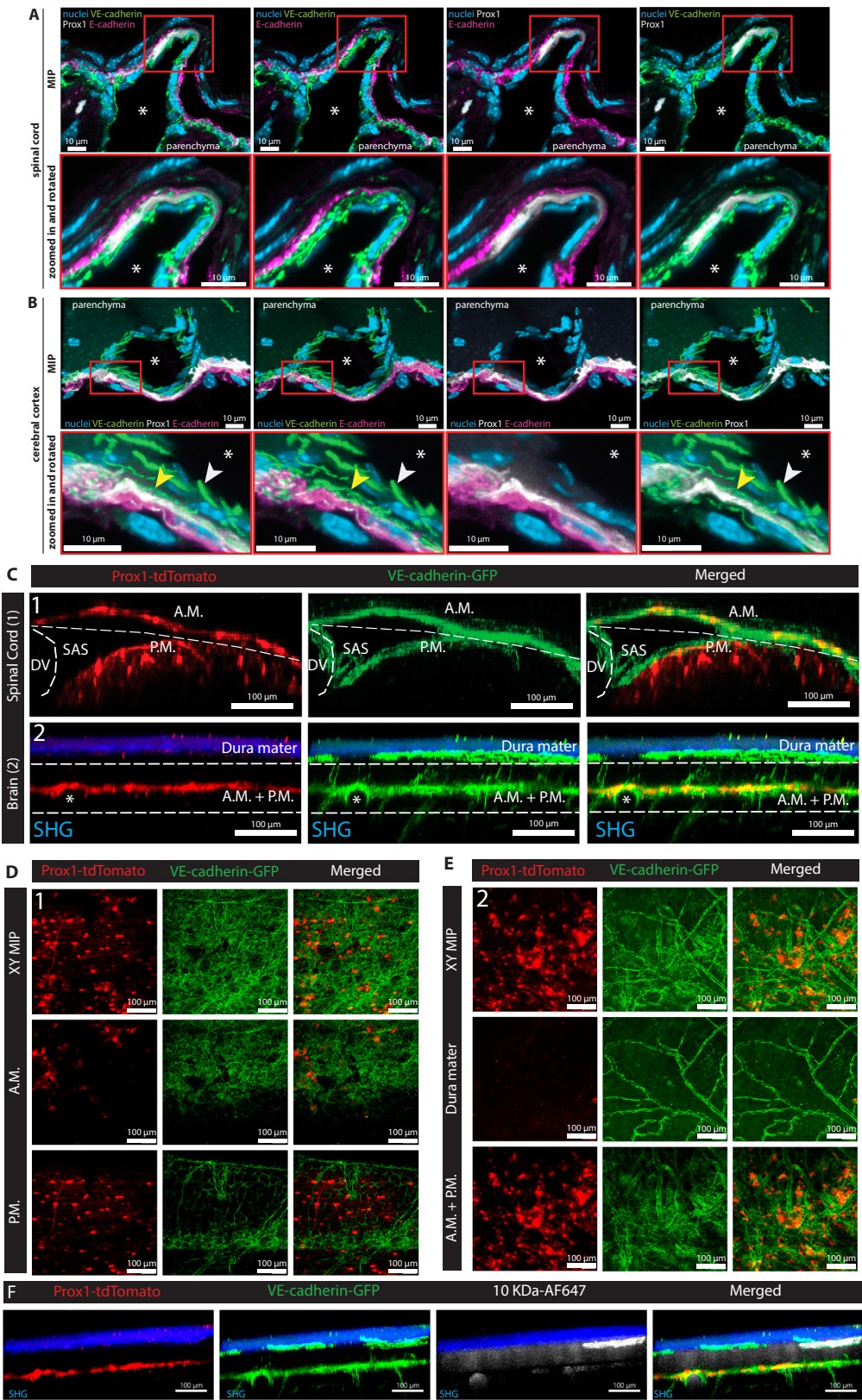

any CD8+ T cells crossing the arachnoid mater into the dura mater. Interestingly, during immune surveillance, slightly more CD8+ T cells were found above than below the pia mater (Fig. 10F). During neuroinflammation, the number of CD8+ T cells per field of view was significantly increased (>10-fold), with most T cells located immediately below the GFP signal from the pial layer, suggesting they reside in the subpial space.

CD8+ T cell-mediated neuroinflammation in the ODC-OVA mouse model is characterized by infiltration of CD8+ T cells into the CNS parenchyma[67,69]. In the absence of a landmark for the glia limitans, we could, however, not determine whether CD8+ T cells were in the subpial space or had breached the glia limitans and infiltrated the superficial CNS parenchyma (Fig. 10D). We, therefore, next asked whether CD8+ T cells would accumulate in the subpial space in

**Fig. 6 | VE-cadherin⁺/Prox1⁺ cells in a layer of arachnoid mater. A, B** Images of thin (10 μm) coronal cryosections of decalcified thoracic spinal cord (**A**) and head (**B**) of VE-cadherin-GFP; Prox1-tdTomato double reporter mice, acquired by confocal laser scanning microscopy. The VE-cadherin-GFP signal (green) is visible in the arachnoid mater, pia mater, and blood vessel endothelial cells. The Prox1-tdTomato signal (white) overlaps the VE-cadherin-GFP signal in the arachnoid. No Prox1-tdTomato signal is evident in the pia mater. E-cadherin immunostaining (magenta) is adjacent to but does not overlap the Prox1-tdTomato signal. White arrowheads mark the VE-cadherin-GFP signal in adherens junctions of vascular endothelial cells. Yellow arrowheads mark the more delicate VE-cadherin-GFP signal in adherens junctions of the leptomeninges. White asterisks mark vessel lumens. Decalcified heads were cut ~2.5 mm caudal to the bregma. Scale bars = 10 μm. **C–E** Images of 2P-IVM of VE-cadherin-GFP; Prox1-tdTomato double reporter mice made through a cervical spinal cord window (**C, D**) or after skull thinning (**C, E**). VE-cadherin-GFP

(green) at junctions is visible in the arachnoid and pia mater and in blood vessel endothelial cells over the brain and spinal cord. Prox1-tdTomato (red) is visible in the arachnoid mater in both locations but in the pia mater only over the spinal cord. The second-harmonic generation signal (blue) in skull thinning preparation (**C**) shows remnants of skull bone and dura mater. **C** YZ MIP of the cervical spinal cord and brain. **D, E** XY MIP of the cervical spinal cord (**D**) and skull thinning (**E**) preparation. XY MIP from full 100-μm Z-stacks are shown in the first row. Spinal cord (**D**) and brain (**E**) cross-sections cropped to show two distinct regions defined by dashed lines in **C. F** YZ MIP images of brain in a skull thinning preparation of a VE-cadherin-GFP; Prox1-tdTomato mouse injected intravenously with 10kDa-AF647-Dextran (white). Representative of images from three different mice. Second-harmonic generation is shown in blue. DV dorsal vein, A.M. arachnoid mater, P.M. pia mater, SAS subarachnoid space; BV blood vessel.

neuroinflammation without crossing the glia limitans into the CNS parenchyma. To this end, we immunostained sections of brain and spinal cord of ODC-OVA; VE-cadherin-GFP knock-in mice for glial fibrillary acidic protein (GFAP) on day 7 after LCMV-OVA infection. We confirmed the presence of CD8⁺ T cells in the SAS (above the VE-cadherin-GFP⁺ cells from the pia mater), in the subpial compartment (between the pial VE-cadherin-GFP⁺ and the GFAP⁺ astrocytes), and within the CNS parenchyma below the GFAP⁺ astrocytes (Fig. 10G).

In neuroinflammation, we also observed slightly more flattened CD8⁺ T cells potentially performing transcellular and paracellular diapedesis across the pia mater, whereas the number of CD8⁺ T cells interacting with the pia mater facing the SAS was not significantly different from immune surveillance in the normal CNS.

Finally, to characterize the motility of CD8⁺ T cells in the different compartments bordered by the leptomeninges, tdTomato⁺ CD8⁺ T cells were manually tracked (Imaris 9.8 software) above and beneath the pia mater. During normal CNS immune surveillance, no differences in the motility of activated CD8⁺ T cells located above versus below the pia mater were observed. However, both the CD8⁺ T cell crawling speeds above and below the pia mater were significantly reduced in neuroinflammation than in normal immune surveillance. Interestingly, in neuroinflammation the crawling speed and displacement of CD8⁺ T cells located under the pia mater was significantly greater than CD8⁺ T cells crawling on the SAS-facing side of the pia mater (Fig. 10H).

Taken together, visualization of the arachnoid mater and pia mater in VE-cadherin-GFP reporter mice in combination with soluble CSF tracers or immune cells enabled precise assignment of cellular localization in relation to the SAS and pia mater. This attribute made it possible to probe the barrier properties of the arachnoid and pia to immune cells and soluble immune mediators.

## Discussion

The meninges are receiving greater attention as a site for CNS immune surveillance (summarized in ref. 70). Multiple recent concepts relevant to CNS immunity suggest free mixing of parenchymal interstitial fluid (ISF) with CSF, according to the glymphatic hypothesis[71], migration of immune cells from the SAS to dural lymphatics[5,72], or migration of immune cells from the dura mater into the SAS or CNS parenchyma[27–30]. These concepts are based predominantly on findings from intravital microscopy (IVM). However, a major drawback of existing IVM techniques is the lack of concurrent visualization of CNS barriers and immune components. This limitation prevents the acquisition of direct localization of immune cells and mediators in relation to specific barriers and other CNS compartments. More sophisticated surgical protocols, advanced stable imaging systems, and novel transgenic fluorescent reporter mouse models would help to achieve this goal. The availability of fluorescent reporter mice suitable for visualizing individual layers of the meninges and barriers in the mouse brain and spinal cord by IVM would be a step toward this goal.

While using our well-established cervical spinal cord window model[38] for 2P-IVM in VE-cadherin-GFP knock-in mice, we made the surprising finding that the arachnoid and pia mater are connected by VE-cadherin-GFP-containing junctions. Using skull thinning and acute cranial window preparations, we confirmed that leptomeningeal cells of the brain of living mice are connected by VE-cadherin-GFP-containing junctions.

VE-cadherin (cadherin-5) is a member of the cadherin family that, for decades, has been considered an adhesion molecule that forms AJs specifically between endothelial cells[73]. As one of the most studied cadherins, VE-cadherin in endothelial cells is known to play a central role in maintaining endothelial cell barrier function and permeability, modulating angiogenesis, immune cell diapedesis, and other vascular functions[47,73].

Optimal adhesive interaction of VE-cadherin molecules in AJs requires association with cytoplasmic catenins. Immunostaining of the brain and spinal cord confirmed the association of α- and β-catenin with VE-cadherin-GFP in endothelial cells and also revealed their association with VE-cadherin-GFP in cells of the arachnoid and pia mater, consistent with the contribution of VE-cadherin to AJs in these cells. A novel feature of the leptomeninges, evident by immunostaining and by 2P-IVM, was the punctate pattern of VE-cadherin-GFP, unlike the typical continuous, linear pattern of VE-cadherin-GFP in endothelial cells. In addition to the punctate pattern, VE-cadherin-GFP had a linear or mixed punctate/linear pattern in some leptomeningeal cells.

The differing patterns of VE-cadherin in leptomeningeal cells deserves further work to understand the functional significance. One possibility is that the punctate pattern reflects VE-cadherin at *maculae adherentes*, whereas the linear pattern indicates *zonulae adherentes*[7,57,58]. Although the two forms of AJs are not readily distinguished in TEM images of thin tissue sections that are largely 2-dimensional, both forms were evident in confocal and 2-photon images of thicker specimens. The two forms of AJs could reflect regional differences in the number of individual leptomeningeal layers or in the functional diversity of cells that form them[74].

Recent transcriptome profiling studies combined with histological and functional studies have provided evidence that the leptomeninges are composed of fibroblast-like cells[75]. Due to the lack of a specific marker(s), the precise phenotype and function of these meningeal cells remain to be described and they may often be confused with pericytes, epithelial cells, or other cell types, due to overlapping cellular markers[75]. In this context, it is interesting to note that the mouse brain atlas (http://mousebrain.org) has previously identified the expression of Cdh1 (E-cadherin) and Cdh5 (VE-cadherin) in a cluster of cells referred to as "ABC cells" and assigned these as "vascular leptomeningeal cells", likely due to the expression of VE-cadherin. A recent single-cell transcriptome profiling study of isolated vascular and perivascular cells of the mouse brain also identified expression of Cdh5 in a cluster outside the endothelial cell clusters and assigned Cdh5 expression to "fibroblast-like cells" due to their

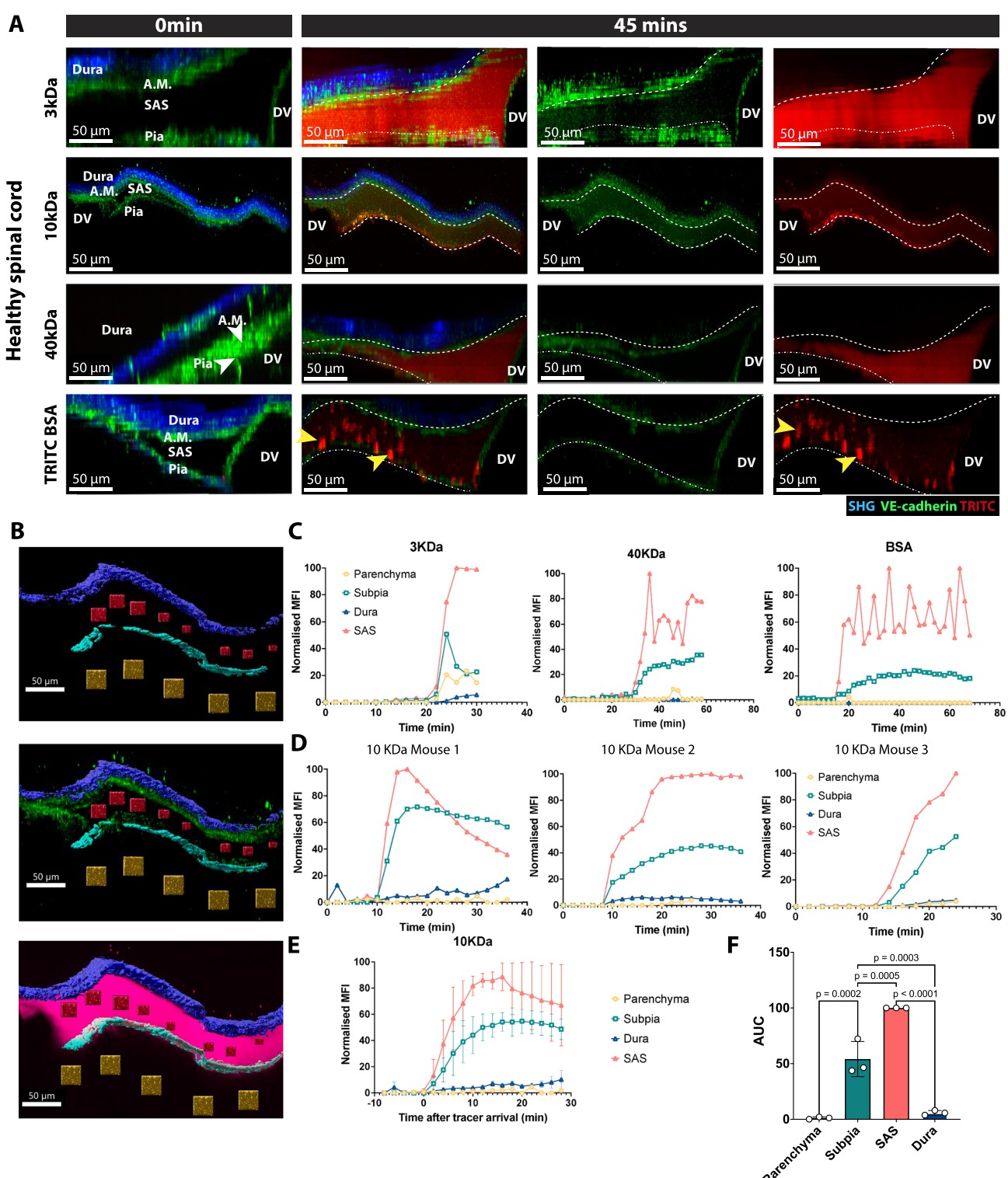

expression profile (http://betsholtzlab.org/VascularSingleCells/database.html)[76]. By combining single-cell transciptomics with multi-modal high-resolution imaging, a recent study identified multiple fibroblast subtypes composing the pia mater, inner arachnoid, arachnoid barrier, and dural border cells[10]. The differential gene expression profiles characterizing these fibroblast subtypes include junctional proteins with VE-cadherin expression being detected at different levels in fibroblasts of the leptomeninges and in dural border cells[10]. Combined with our present observations of VE-cadherin-GFP in the cellular junctions of the inner arachnoid and pia mater, these data support the interpretation that VE-cadherin is expressed in fibroblast-like cells of the arachnoid and pia mater in the mouse.

We observed that VE-cadherinGFP+ meningeal layers covered all surfaces of the CNS, but found morphological differences between the brain and the spinal cord. The observed morphological differences may be due to the distinct developmental origins of the meningeal layers of the brain and spinal cord[42,43]. The large web-like appearing VE-cadherin-GFP+ spinal cord pial and arachnoid cells resemble exactly the pattern of cells described in the classic work of Key and Retzius[77], where after silver staining of the outer layers of the spinal cord of dogs

**Fig. 7 | Visualization of CNS zonation created by leptomeningeal barriers in healthy VE-cadherin GFP reporter mice in vivo. A** Representative images of the spinal cord window 2P-IVM imaging of a healthy VE-cadherin-GFP knock-in reporter mice. Before the spinal cord window preparation, a tracer-filled cannula was implanted into the cisterna magna. During 2P-IVM, the mice were infused with 2.5 µl of either 3 kDa, 10 or 40 kDa TRITC dextran, or TRITC BSA at a rate of 1 µl/min using a syringe pump. XY time-lapse sequence of a 400 µm × 400 µm scan field at a depth of 160–220 µm and 81–111 z-projections with 2 µm spacing were acquired for 45 mins (spinal cord). The dura mater is visible in blue due to the second-harmonic generation of the collagen type 1 fibers in the dura. Arachnoid mater (A.M.) and pia mater are visible in green due to the VE-cadherin-GFP expression. Infused tracer is seen in red. YZ MIP of the meningeal layers of the spinal cord of a healthy VE-cadherin GFP knock-in reporter mouse is shown. At 0 min, no tracer (red) was seen. At 45 mins, the 3KDa, 10 kDa, and 40 kDa TRITC dextran and TRITC BSA (red) crossed the pia mater but not the arachnoid mater. Yellow arrowheads highlight phagocytic cells in SAS that have taken up BSA. Images are representative of three independent experiments per tracer. **B** YZ MIP of the spinal cord meningeal layers of a VE-cadherin GFP knock-in reporter mouse showing the segmented volumes for longitudinal quantification of the fluorescence intensity of the cisterna magna injected TRITC tracers (magenta). Dura mater (dark blue) and subpial compartment (light blue) are segmented based on the second-harmonic generation signals. SAS (red) and spinal cord parenchyma (dark yellow) are segmented as cubes distributed in between the two VE-cadherin-GFP layers (green) or 20–50 µm under the VE-cadherin-GFP+ pia mater, respectively. **C** VE-cadherin-GFP knock-in mice were infused with 3-, 40-kDa-TRITC, and TRITC-BSA tracers into the cisterna magna and 2P-IVM was performed over time. Graphs show the longitudinal quantification of the mean fluorescence intensity of the injected tracer in the segmented volumes from the spinal cord meningeal cross-section shown in **B**. Data are normalized to the highest MFI value detected in the SAS after background signal subtraction. Background signal was determined as the average fluorescence signal measured in all segmented volumes prior to tracer injection. Each graph shows the full quantification of one mouse. Source data are provided as a Source Data file. **D** VE-cadherin-GFP mice were injected via the cisterna magna with 10-kDa-TRITC tracers and 2P-IVM was performed over time. Graphs show the longitudinal quantification of the mean fluorescence intensity of the injected tracer in the segmented volumes from the spinal cord meningeal cross-section shown in **B**. Data are normalized to the highest MFI value detected in the SAS after background signal substruction. Background signal was determined as the average fluorescence signal measured in all segmented volumes prior to tracer injection. Each graph shows the quantification of one individual mouse. Source data are provided as a Source Data file. **E**, **F** Graphs in **D** were combined after matching the dynamics of tracer arrival in the SAS of the field of view (FOV) (**E**). Area under the curve (AUC) was calculated in the different segmented volumes for the first 12 min after tracer arrival in the FOV. Data were pooled from three independent experiments and are shown as mean ± SD. Source data are provided as a Source Data file. **F** Data were pooled from three independent experiments and analyzed using one-way ANOVA with Tukey's multiple comparisons test. Data are shown as mean ± SD. Source data are provided as a Source Data file. DV dorsal vein, A.M. arachnoid mater.

and rabbits, they described delicate endothelial pellicles (Endothelhäutchen). Similar observations were obtained almost 100 years later by staining of the cellular borders of isolated leptomeninges from the brain of rats[78]. These authors termed these cells "squamous epithelial cells" of the leptomeninges. Based on the resemblance of the morphology of these previously described flat and delicate cellular layers of the leptomeninges, we speculate that these are the "endothelial-like" cellular layers that are the VE-cadherin-GFP+ cells observed in our 2P-IVM images.

The arachnoid mater is composed of several layers of cells, with the outermost cells forming the arachnoid barrier. These arachnoid barrier cells are connected by TJs formed by claudin-11 and AJs formed by E-cadherin in rodents and humans[14,53,79,80]. Arachnoid barrier cells furthermore express transporters and efflux pumps comparable to the endothelial cells that form the BBB and to choroid plexus epithelial cells that form the blood-CSF barrier in brain ventricles[13,16].

We found that most of the VE-cadherin-GFP signal did not overlap E-cadherin immunoreactivity in arachnoid barrier cells. However, a small subset of arachnoid barrier cells co-expressed VE-cadherin and E-cadherin. Expression of Cdh5 has indeed been found in Cdh1+ arachnoid barrier cells at fetal stages in mice[81] and is maintained during adulthood[10]. As cadherins mediate strictly homophilic adhesions, some arachnoid barrier cells form two or more molecularly distinct AJs that may have different functions. The continuity of the VE-cadherin-GFP signal in the arachnoid mater and trabeculae into the pia mater of VE-cadherin-GFP knock-in mice suggests that these VE-cadherin expressing fibroblast-like cells face the CSF-filled SAS and are distinct from arachnoid barrier cells.

A recent study proposed that a layer of Prox1-expressing mesothelial cells, designated "subarachnoid lymphatic-like membrane (SLYM)", forms a barrier that divides the SAS into upper and lower compartments[59]. To test this claim, we crossed VE-cadherin-GFP knock-in mice with Prox1-tdTomato reporter mice to determine the location of the Prox1+ layer. Confocal microscopic and 2P-IVM imaging of spinal cord and brain of these double reporter mice (VE-cadherin-GFP; Prox1-tdTomato mice) revealed that Prox1-tdTomato+ meningeal cells were part of the VE-cadherin-GFP+ cellular layers of the arachnoid mater and that the VE-cadherin-GFP+/Prox1-tdTomato+ cells reside directly below the E-cadherin+ arachnoid barrier cells. We also found that Prox1-tdTomato+ cells did not form an entirely continuous layer, a feature that is inconsistent with the putative barrier function of these cells.

In studies designed to identify the location of the barrier, we found that intravenously injected fluorescent tracers extravasated from fenestrated blood vessels in the dura mater but were blocked from entering the SAS at the level of the VE-cadherin-GFP+ arachnoid mater by the arachnoid barrier cells. The limited Z-axis resolution of 2P-IVM hampers differentiation of a potential barrier property of the VE-cadherin-GFP+/Prox1-tdTomato+ cells from that of the directly adjacent arachnoid barrier cell layer. Thus lack of simultaneous visualization of the Prox1-expressing meningeal cells with the arachnoid mater in this previous study[59] prevented a verifiable positioning of the Prox1-expressing cells within the SAS leading to the erroneous conclusion of the existence of a 4th meningeal membrane. VE-cadherin-GFP and VE-cadherin-GFP; Prox1-tdTomato reporter mice can now be used with E-cadherin and claudin-11 immunoreactivities in future studies to identify key cellular components responsible for barrier functions and other functional properties of the meninges.

Infusion of fluorescent tracers of different molecular weights into the SAS at the cisterna magna of VE-cadherin-GFP knock-in mice enabled testing of the barrier properties of the arachnoid and pia mater of healthy mice. By quantifying the dynamics of tracer signal enhancement in the SAS, dura, subpial space, and parenchyma of the spinal cord, we were able to assess the barrier properties of the leptomeninges by in vivo imaging, despite the limited resolution of 2P-IVM in the z axis. The arachnoid was found to be a barrier even for low molecular weight tracers of 3 kDa. At the same time, the pia mater allowed for passage of all tracers tested, small, ranging from 3 kDa to 40 kDa Dextran and 66 kDa BSA. Our observations thus suggest that chemokines (8–10 kDa)[82] and small peptides like the MOG[83] can readily cross the pia mater but not the arachnoid mater, underscoring tight barrier properties of this outer layer of the leptomeninges.

The arachnoid mater also is a barrier for molecules derived from outside of the brain, as previously shown by Balin and colleagues who observed that peripherally injected horseradish peroxidase fails to move from the dura mater across the arachnoid layer into the SAS[3]. In accordance with these findings, we here observed that systemically injected 10 kDa dextran is prevented by the arachnoid mater from moving from the dura mater to the SAS.

The arachnoid mater thus forms a barrier to molecules moving in both directions between the dura mater and the SAS. Taken together,

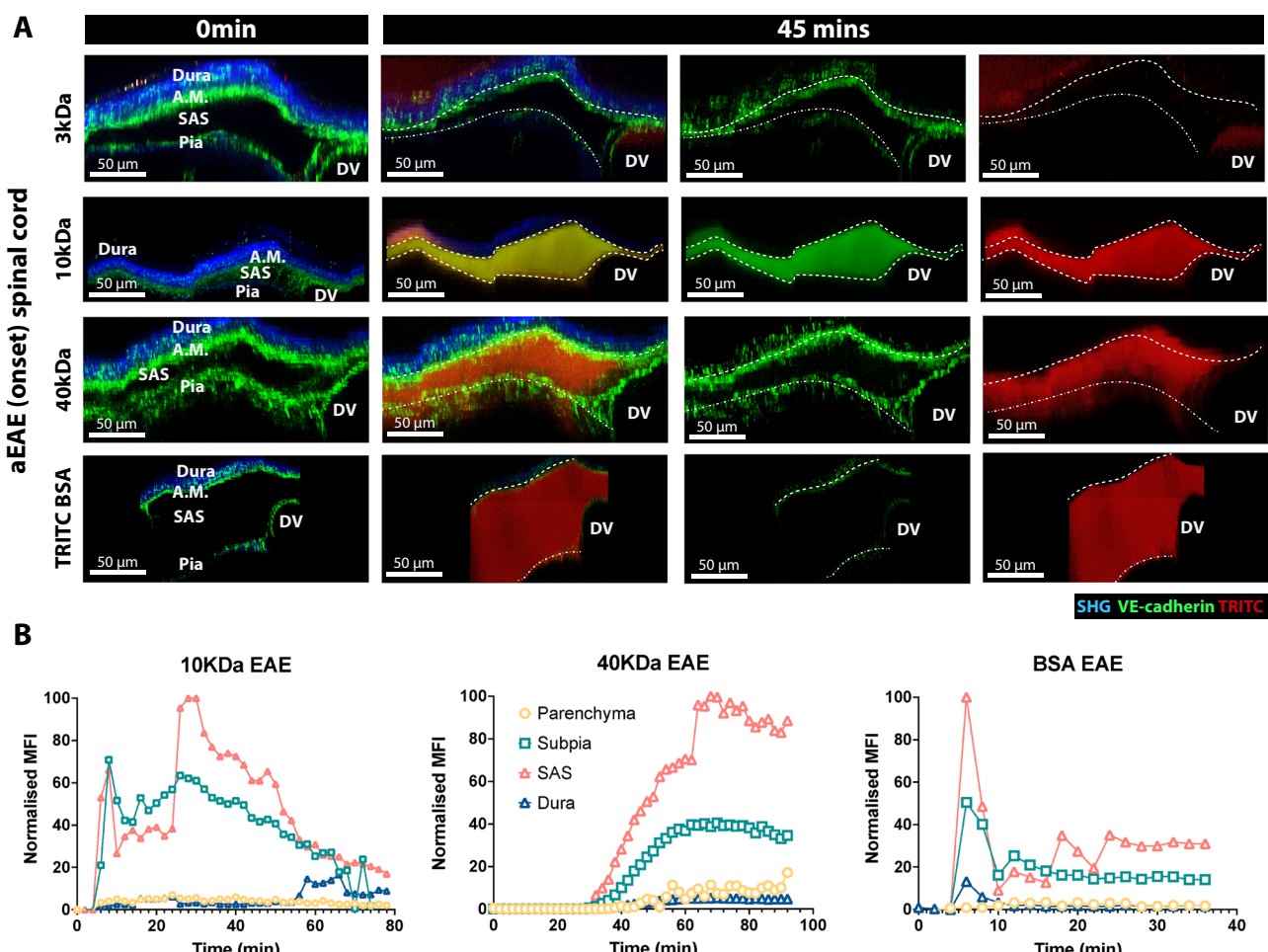

**Fig. 8 | The VE-cadherin-GFP reporter mouse allows for in vivo visualization of alterations in leptomeningeal CNS zonations during neuroinflammation.**
**A** Representative images of the spinal cord window 2P-IVM imaging of VE-cadherin-GFP knock-in reporter mice suffering from aEAE. Before a spinal cord window preparation, a tracer-filled cannula was implanted into the cisterna magna. During 2P-IVM, the mice were infused with 2.5 µl of either 3 kDa or 10 kDa, or 40 kDa TRITC dextran or TRITC BSA at a rate of 1 µl/min using a syringe pump. XY time-lapse sequence of a 400 µm × 400 µm scan field at a depth of 160–220 µm and 81–111 z-projections with 2 µm spacing were acquired for 45 mins. The dura mater is visible in blue due to the second-harmonic generation of the collagen type 1 fibers in the dura. Arachnoid mater (A.M., highlighted by a dashed line) and pia mater (highlighted by a dashed/dotted line) are visible in green due to the VE-cadherin expression. Injected tracer is seen in red. YZ MIP of the meningeal layers of the spinal cord of a VE-cadherin GFP knock-in reporter mouse suffering from aEAE imaged at the onset of the disease (days 13–15 p.i., clinical score +) are shown. At 0 min, no tracer (red) is seen. At 45 min, the 3KDa TRITC dextran (red) is seen in the dorsal vein (DV) and above the dura mater (blue), The 10- and 40-kDa TRITC dextran as well as the TRITC-BSA tracer (red) crossed the pia mater but not the arachnoid mater. Images are representative of three mice per tracer. **B** Graphs show the longitudinal quantification of the mean fluorescence intensity of the injected tracer in the segmented volumes from the spinal cord meningeal cross-sections shown in Fig. 7B. Data are normalized to the highest MFI value observed in the SAS after background signal substraction. Background signal was determined as the average of fluorescence signals observed in all segmented volumes prior to tracer injection. Each graph shows the full quantification of one mouse. Source data are provided as a Source Data file.

these observations raise questions about the interpretation of current concepts, based on studies that omit the barrier properties of the arachnoid mater, that advocate unrestricted communication between chemokines and CNS parenchymal-derived antigens with immune cells in the dura mater as they omit the barrier properties of the arachnoid mater[28,29,36]. Further studies are thus needed to determine the barrier properties of the arachnoid mater to immune mediators such as cytokines, chemokines, or antibodies.

Making use of the VE-cadherin-GFP landmark for the leptomeninges during neuroinflammation allowed us to determine volume changes of the SAS in the brain and spinal cord and gave insight into possible alterations of barrier properties of the arachnoid and pia mater. The findings support the notion that BBB breakdown at the onset of aEAE affects the SAS, while the barrier properties of the leptomeninges were not impaired at this early timepoint. Later stages of aEAE may lead to alterations in leptomeningeal barrier properties that further impact CSF flow and the distribution of immune cells and mediators in CNS compartments regulated by these barriers. Combined impairment of multiple brain barriers will impact CNS communication with the peripheral immune system and ultimately contribute to disease pathogenesis (summarized in ref. 62).

Following the trafficking of circulating T cells into the CNS of VE-cadherin-GFP knock-in mice, we observed T cells interacting with the inflamed BBB endothelium lining the spinal cord microvessels in accordance with previous observations by us and others[38,84]. The molecular mechanisms involved in the multi-step T cell extravasation across the BBB have been described as involving the sequential interaction of adhesion and signaling molecules on the surface of T cells and on the luminal surface of BBB endothelial cells (summarized in ref. 85). Here, we observed that activated T cells that had reached the SAS crawl on the pia mater. This finding provides evidence that the pia mater serves as a barrier for immune cells. Therefore, T cells have to find a site permissive for diapedesis across the pia comparable to diapedesis across endothelium (summarized in ref. 85).

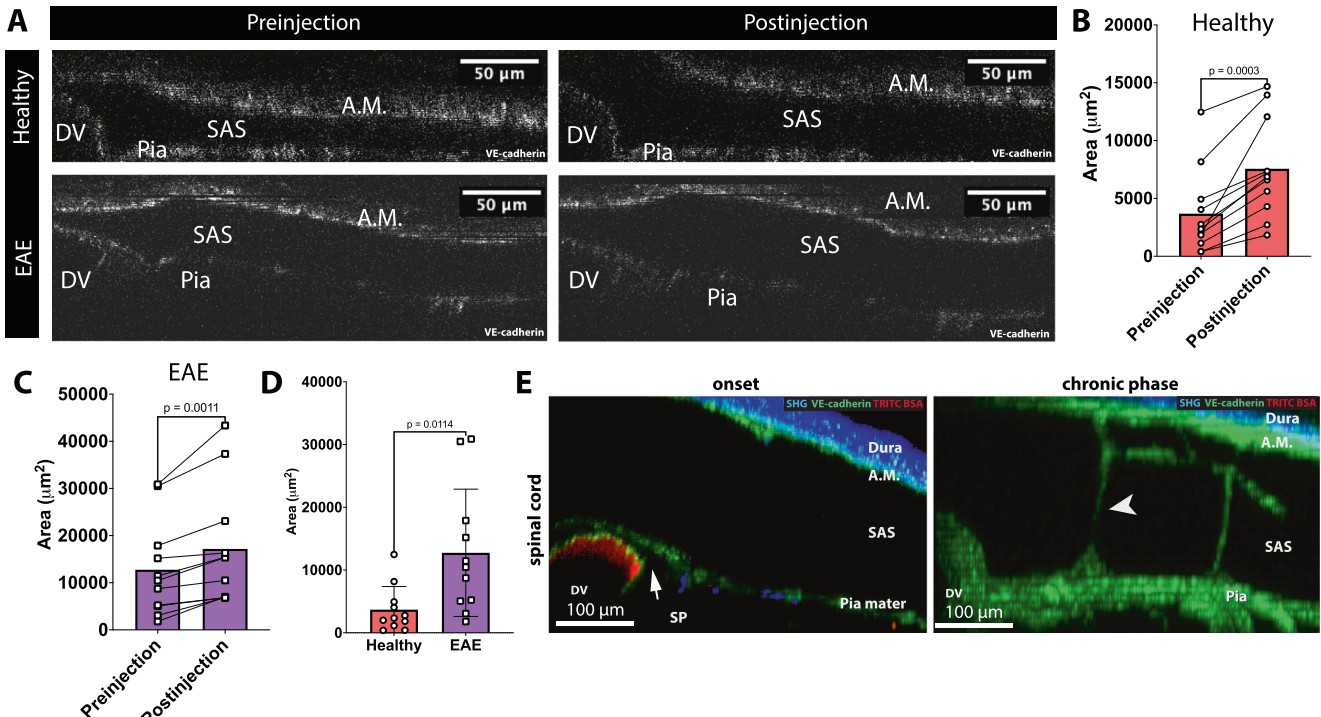

**Fig. 9 | The VE-cadherin-GFP knock-in mouse allows for visualization of dynamic alterations of CSF spaces in health and under neuroinflammation.**
**A** Representative cross-section images of the spinal cord meninges from healthy VE-cadherin GFP reporter mice and VE-cadherin-GFP mice suffering from aEAE at the onset of disease prior and post cisterna magna injection of tracers. The SAS was defined as the space bordered by VE-cadherin GFP signal (white) from the arachnoid mater (A.M.), the pia mater and from the endothelial cells of the dorsal vein (DV). Images are representative from 11 independent experiments per condition. **B–C** Quantification of the SAS area in cross-section images of the spinal cord meninges from healthy. **B** VE-cadherin-GFP reporter mice and VE-cadherin-GFP mice suffering from aEAE at the onset of disease (**C**) prior and post cisterna magna injections of tracers. SAS area is defined as the space bordered by VE-cadherin GFP signal (white) from the arachnoid mater (A.M.), pia mater and from the endothelial cells of the dorsal vein (DV). Data were pooled from 11 independent experiments and analyzed using two-sided paired parametric T-test, and are represented as mean ± SD. Source data are provided as a Source Data file. **D** Quantification of the SAS area in cross-section images of the spinal cord meninges from healthy VE-

cadherin-GFP reporter mice and VE-cadherin-GFP mice suffering from aEAE at the onset of disease prior cisterna magna injections of tracers. SAS area is defined as the space bordered by VE-cadherin GFP signal (white) from the arachnoid mater (A.M.), pia mater, and from the endothelial cells of the dorsal vein (DV). Data were pooled from 11 independent experiments per condition and analyzed using two-sided parametric *T* test, and are represented as mean ± SD. Source data are provided as a Source Data file. **E** Representative images of the spinal cord window 2P-IVM imaging of a VE-cadherin-GFP knock-in reporter mouse suffering from EAE at the onset (day 14 p.i., clinical score +) and at the chronic phase (day 25 p.i., clinical score +). The dura mater is visible in blue due to the second-harmonic generation of the collagen type 1 fibers in the dura. VE-cadherin-GFP is visible in green. Fluorescent Dextran was injected intravenously to visualize the blood vessels (red). At both time points, the dura mater (blue), arachnoid mater (A.M.), and pia mater are seen. A large SAS with trabeculae (white arrowhead) between the arachnoid mater and pia mater is visible. At the onset timepoint, a subpial space (SP) is visible between the dorsal vein (DV) and the pia mater (highlighted by a white arrow). Images are representative of seven mice per condition.

Entry of effector T cells into the SAS of the spinal cord has been observed in a Lewis rat aEAE model[84]. 2P-IVM identified T cells in the SAS to move along collagen fibers and eventually to be flushed away by CSF with some T cells able to reattach to the leptomeninges. This earlier study also identified T cell integrins α4β1 (VLA-4) and αLβ2 (LFA-1) as mediating T cell interaction with leptomeningeal cells, which is reminiscent of the molecular mechanisms involved in T cell interaction with BBB endothelial cells. In accordance with these previous findings, our present observations confirm that VE-cadherin-GFP⁺ cells of the pia mater present a permissive barrier to T cells. The molecular mechanisms mediating the multi-step migration of T cells across the pia mater under conditions of CSF flow need further investigation but could serve as a selection process for T cells "allowed" to enter the CNS parenchyma, where they could have beneficial or detrimental effects.

By visualizing the leptomeningeal layers in VE-cadherin-GFP reporter mice by 2P-IVM, we identified some of the limitations of this technique that requires making surgical windows and tracer injections. One limitation is that artifactual or expanded spaces formed in some preparations, where an apparent "subdural" space appeared during the skull thinning but not when making an acute cranial window preparation. There is general agreement that under physiological

conditions cranial meninges do not harbor a subdural space[8,86]. However, such a subdural space can form by detachment of dural border cells as a result of tissue damage after surgery, can be caused by a hemorrhage leading to subdual hematoma, or can occur after death at autopsy[7,8]. While we do not yet understand the mechanisms causing the apparent "subdural" space specifically after skull thinning, by allowing for simultaneous imaging of the meningeal layers, the VE-cadherin-GFP reporter mouse will be very suitable to improve the surgical setups required for 2P-IVM of the CNS.

Another limitation is that the volume of the spinal cord SAS space can rapidly increase during even low rates of tracer infusion at the cisterna magna. When the leptomeninges are not visible during 2P-IVM, these changes can be missed and lead to erroneous localization of immune cells and mediators in the CNS.

A further limitation is that the Z-axis resolution of 2P-IVM, even with our advanced imaging stabilization protocol VivoFollow 2 in place[41], does not permit imaging or measurement of tracer or cell movement through the thin cell layers of the leptomeninges. However, this limitation can be circumvented by monitoring the presence or absence of fluorescent cells or molecules on either side of the VE-cadherin-GFP⁺ arachnoid or pia mater. Nevertheless, examination of

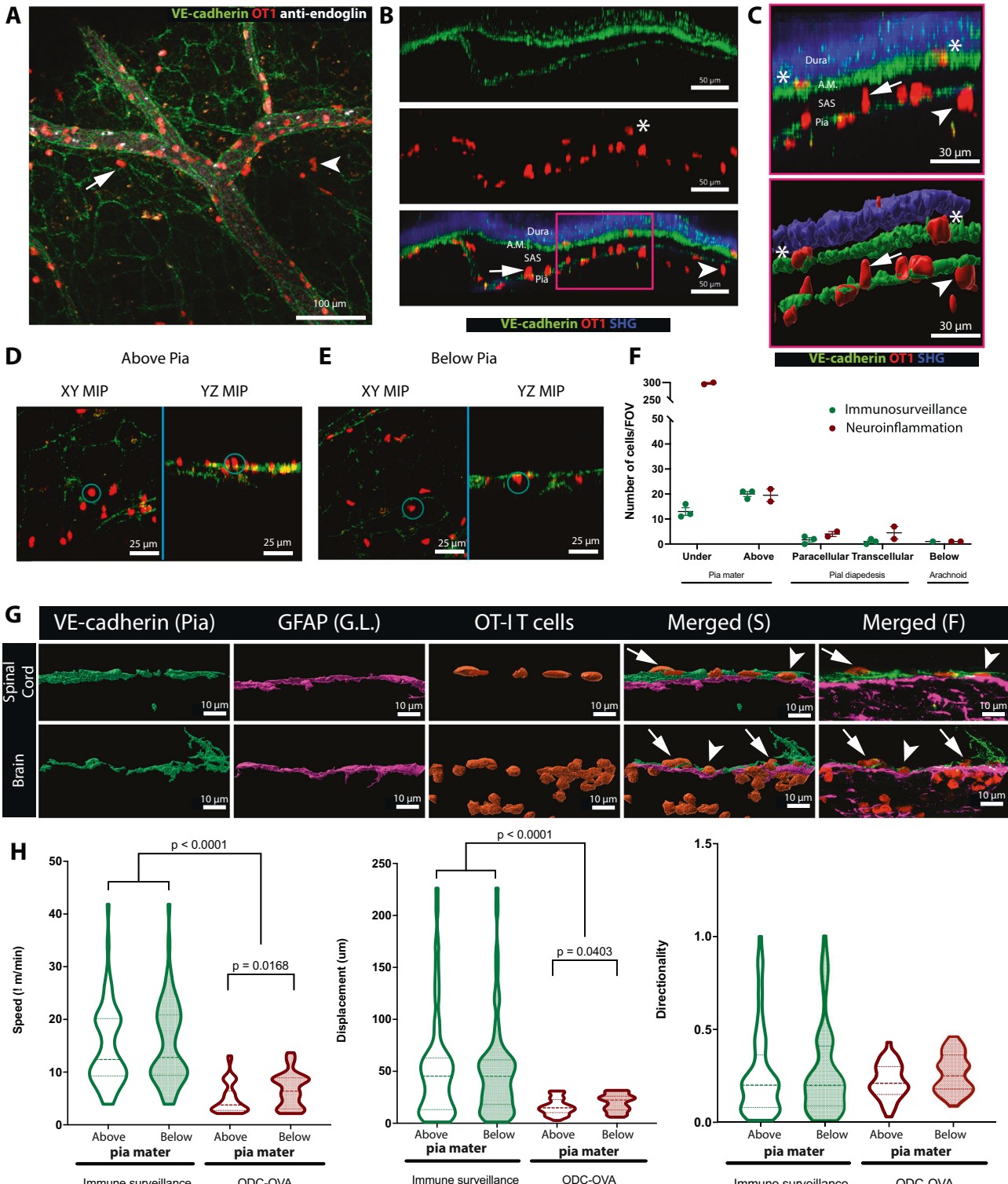

the mechanisms of how T cells cross the pia mater cells in vivo will continue to be challenging without further technical advances.

We conclude that our findings collectively provide multiple lines of evidence showing that fibroblast-like cells of the arachnoid and pia mater express VE-cadherin and form VE-cadherin⁺ AJs distinct from those in endothelial cells. The data also show that VE-cadherin-GFP knock-in mice are a powerful tool for in vivo imaging of CNS leptomeningeal layers. These mice enable direct visualization of CNS compartments that differ in their accessibility to immune cells and mediators (summarized in Supplementary Fig. 6). By taking advantage

of the vast knowledge of VE-cadherin in endothelial cells, our study sets the stage for advancing the understanding of the functions of VE-cadherin in leptomeningeal cell development, spatial organization, and barrier formation and maintenance. In addition, VE-cadherin-GFP knock-in mice facilitate imaging of the distribution of cellular and humoral immune responses in the CNS, both in health and in disease. Thereby, these mice make it possible to exploit anatomical landmarks that delineate the distribution of CSF and immune cells and mediators that enter the CSF, and thus advance the understanding of mechanisms that maintain CNS immune privilege.

**Fig. 10 | The VE-cadherin GFP reporter mouse allows for visualization of immune cell trafficking across the leptomeninges in vivo during health and neuroinflammation.** Representative images of spinal cord window 2P-IVM imaging of a VE-cadherin-GFP knock-in reporter mouse suffering from a CD8[+] T-cell-mediated neuroinflammatory disease. On day 7 after viral infection, the mice were injected intravenously with a fluorescently labeled anti-endoglin antibody (white) to highlight the blood vessel lumen (BV). The dura mater is visible in blue due to the second-harmonic generation of the collagen type 1 fibers in the dura. OT-I CD8[+] T cells are depicted in red. **A** X−Y MIP of the pia mater. White arrowhead shows an OT I CD8[+] T cell (red) that has crossed the pia mater, and white arrow highlights one above the pia mater. See also Supplementary Movie 9. Data are representative of ten independent experiments. **B, C** YZ MIP of the meningeal layers of the spinal cord. The arachnoid mater (A.M.) and pia mater are visible in green due to VE-cadherin-GFP expression. OT I CD8[+] T cells (red) are visible inside the blood vessel (BV), in the SAS, above and below the pia mater. White asterisk indicates a CD8[+] T cell in the arachnoid mater. **C** Zoom-in of the magenta box in **B** showing the fluorescence signals (top panel) and the segmented surfaces (bottom panel) of the dura mater, arachnoid mater, pia mater, and OT I CD8[+] T cells. Surfaces were rendered with Imaris 9.8 software. White arrow points to an OT I CD8[+] T cell above the pia mater, white arrowhead points to an OT I CD8[+] T cell below the pia mater and asterisks point to OT I CD8[+] T cells interacting with arachnoid mater. White asterisks indicate CD8[+] T cells in the arachnoid mater. Data are representative of seven mice imaged in seven different experiments. **D, E** XY and YZ MIP images of OT I CD8[+] T cells (red) localized right above (**D**) or below (**E**) pia mater (green) in a VE-cadherin-GFP knock-in mouse. See also Supplementary Movie 10. Data are representative of three independent experiments. **F** Quantification of the number of OT I CD8[+] T cells observed in different CNS compartments during immune surveillance and neuroinflammation. Data are pooled from three control mice ("immunosurveillance") and 2 under neuroinflammation and shown as mean values. Source data are provided as a Source Data file. **G** Brains and spinal cords from ODC-OVA; VE-Cadherin-GFP knock-in mice were harvested on day 7 after induction of autoimmune neuroinflammation. GFAP immunofluorescence staining was performed on 20 µm cryosections. Images show the surface segmentations of the signals of VE-cadherin-GFP at the pia mater (green), glia limitans (purple), and OT I CD8[+] T cells (red) rendered with Imaris 9.8 software. Original fluorescence images are shown on the right panel. White arrows point to OT-I CD8[+] T-cell above the pia mater, white arrowhead points to an OT I CD8[+] T cell below the pia mater. Images are representative of three independent experiments. **H** Violin plots of the crawling speed (µm/min), displacement, and directionality of OT I CD8[+] T cells (a total of 30 cells per condition were analyzed in immunosurveillance from a total of 3 VE-cadherin-GFP knock-in mice and 30 cells per condition were analyzed during neuroinflammation from a total of two ODC-OVA; VE-cadherin-GFP knock-in mice). Cell tracking was performed with Imaris 9.8 software. Data were analyzed using two-sided unpaired parametric T test. Source data are provided as a Source Data file.

## Methods
All our methods comply with the respective ethical regulations. Our animal procedures and experiments have been approved by the Veterinary Office of the Canton Bern (permit numbers BE31/17, BE77/18, and BE98/29).

### Mice
Female mice between 6 and 12 weeks were used in this study. VE-cadherin-GFP knock-in mice expressing a C-terminal GFP fusion protein of VE-cadherin in the endogenous VE-cadherin locus were provided by Prof. Dietmar Vestweber (Münster, Germany)[37,40]. C57BL/6 J wild-type mice were obtained from Janvier (Genest Saint Isle, France). OT-I tdTomato mice expressing a tdTomato reporter ubiquitously were generated by crossing OT I mice (C57BL/6J-Tg (Tcra/Tcrb)1000Mjb) containing a transgenic TCR, recognizing OVA residues 257−264 (SIINFEKL) in the context of H2K[b87], with Ai14 tdTomato reporter mice (Gt (ROSA)26Sor[tm14(CAG-tdTomato)Hze]), in which the Stop-cassette had previously been deleted by breeding with ZP3-Cre transgenic (Tg (Zp3-cre) 93Knw) mice. Prox1-tdTomato mice (Tg(Prox1-tdTomato)TA76Gsat[88]) were backcrossed into the C57BL/6 J background for at least 10 generations and were subsequently crossed into the VE-cadherin-GFP knock-in mice in order to generate the double reporter VE-cadherin-GFP; Prox1-tdTomato mice.

ODC-OVA mice have been described before[67] and were subsequently backcrossed to the C57BL/6 J background for at least 10 generations. ODC-OVA; VE-cadherin GFP knock-in mice were generated by crossing the VE-cadherin-GFP knock-in mice described above with the ODC-OVA mice.

Mice were housed in individually ventilated cages under specific pathogen-free conditions at 22 °C with a 13:11 hr light:dark cycle and free access to water and chow. Animal procedures executed were approved by the Veterinary Office of Canton Bern (permit no. BE31/17, BE77/18, and BE98/20) and are in keeping with institutional and standard protocols for the care and use of laboratory animals in Switzerland.

### Active EAE
Active EAE (aEAE) was induced in 8–12 weeks old female VE-cadherin-GFP knock-in mice in C57BL/6 J background using the MOG$_{aa35-55}$-peptide using a protocol established before[89,90]. Briefly, 200 µg of MOG$_{aa35-55}$-peptide in Complete Freund's Adjuvant (CFA, LabForce; Santa Cruz Biotechnology) supplemented with 4 mg/ml nonviable, desiccated Mycobacterium tuberculosis (H37RA; Difco/BD Biosciences/BD Clontech) were injected subcutaneously on day 0. Pertussis toxin (List Biological Laboratories, Campbell, US) (300 ng in 100 µl PBS/mouse) was applied intraperitoneal (i.p.) on day 0 (immunization day) and day 2 post-immunization. Weights and clinical severity were assessed twice daily and scored as follows[91]: 0, asymptomatic; 0.5, limp tail; 1, hind leg weakness; 2, hind leg paraplegia; 3, hind leg paraplegia and incontinence.

### naive CD8[+] T-cell isolation
Peripheral lymph nodes and spleens from tdTomato+ OT-I C57BL/6 J mice were harvested and single-cell suspensions were obtained by homogenization and filtration through a sterile 100 µm nylon mesh. A second filtration was applied after erythrocyte lysis (0.16 M NH4Cl, 0.17 M Tris-HCl). tdTomato+ OT I cells were isolated with magnetic CD8[+] T-cell selection beads (EasySep, STEMCELL Technologies). The purity of the CD8[+] T cells was assessed by flow cytometry and was >98.5% in each experiment.

### T-cell-mediated CNS immune surveillance and neuroinflammation
Nine- to twelve-week-old VE-cadherin-GFP knock-in mice and ODC-OVA; VE-cadherin GFP knock-in mice were injected intravenously with freshly isolated naive OT I Td Tomato[+] CD8[+] T cells (2 × 10[5] cells in 100 µL/mouse). 24 hr later, mice were peripherally challenged with an intraperitoneal injection of 10[5] plaque-forming units (PFU) of LCMV-OVA[68,69]. Animals were monitored daily for clinical symptoms and scored as follows: 0, healthy; 0.5, partial loss of tail tonus; 1, complete loss of tail tonus; 1.5, hind leg paraparesis; 2, hind leg paraplegia; 3, hind leg paraplegia with incontinence as previously described[68,69].

### General surgery procedures for in vivo imaging
Healthy mice and mice with aEAE were used for imaging experiments at onset (day 14, 15 p.i., clinical score +) or at the chronic phase of the disease (day 25, 35 p.i., clinical score +). Mouse models of CD8[+] T-cell-mediated CNS immune surveillance and neuroinflammation were imaged 7 days post viral infection when clinical score was between 0.5 and 1.

Animals were administered a single dose of fentanyl (0.05 mg/kg)/midazolam (5 mg/kg)/medetomidine (0.5 mg/kg) via intramuscular injection in order to perform a tracheotomy and implant a carotid

catheter (polyurethane 0.2 mm internal diameter, Ref BC-1P, Access technologies, IL) exactly as described before[38]. Arterial blood supply allowed for systemic injection of exogenously labeled cells and fluorochrome-labeled plasma markers or antibodies, thus allowing for immediate imaging as described before[92]. During all surgical procedures and intravital imaging, vital parameters as ECG and body temperature were constantly monitored and recorded. Mice were administered a single dose of buprenorphine (Temgesic; 0.3 mg/kg body weight) subcutaneously to provide adequate pain relief during surgical and imaging procedures and maintained under anesthesia via a tracheotomy using mechanical ventilation (Minivent, Model 845, Harvard Apparatus) with a gas mix of air and oxygen-containing 0.5–1% isoflurane[38].

### Fluorescent tracer infusion into the bloodstream and the cerebrospinal fluid compartment

During 2P-IVM imaging, the anaesthetized surgically prepared mice were systemically injected through the carotid artery with 155 KDa TRITC dextran (2 µg/mouse, Cat #: T128, Sigma-Aldrich, Switzerland) as described to highlight the blood vessels[38]. Where indicated, for tracer infusion into the cisterna magna, healthy anesthetized VE-cadherin-GFP mice were implanted with a tracer-filled cannula in the cisterna magna (CM) as previously described[93]. Briefly, the dura mater was exposed by dissection of the neck muscles that allowed the insertion of a cannula (Ref. BC-1P, Access technologies, IL) mounted onto a cut 30 G needle (Omnican 50, B.Braun) inserted into the cisterna magna and sealed using tissue glue (VetBond™, 3 M) prior to a cranial or spinal cord window preparation. The cannula was then mounted on another 30 G syringe equipped with a longer catheter (SCI: Ref. BB31695-PE/1) and fixed on an syringe pump (Stoelting, Wood Dale, IL). During 2P-IVM, infusion of 2.5 µl of either 3KDa TRITC dextran (Cat #: D3307), 40-KDa TRITC dextran (Cat #: D1842), or TRITC BSA (Cat #: A23016) at a concentration of 20 mg/ml was performed at a speed of 1 µl/min using a syringe pump. Images were acquired every 70 s for 70 min.

### Cervical spinal cord and cranial window/skull thinning preparations

For all preparations, the surgery area was shaved and sterilized. The cervical spinal cord window preparation allowing of 2P-IVM of cervical spinal cord at the level of C3-C4 was performed by a methodology, which has been described in detail as method paper by us before[38]. In brief, animals were anesthetized and prepared as described above. After the preparation of the animal for maintenance of vital parameters, mice were placed on prone position and fixed in a stereotactic frame. A skin incision was introduced from the neck to the bottom of the shoulder plate and muscle covering the spine was opened and fixed to the side of the stage using suture wire. The spine was cleaned of all muscles and a laminectomy was performed from C2 to C6 under a stereomicroscope.

For brain imaging we used both, acute cranial window preparations and skull thinning as each preparation has its own limitations as previously described[94,95]. The acute cranial window preparation was adapted from previously published protocols[46,96], and skull thinning was performed as previously described by Christie et al.[97]. Briefly, for both preparations, local analgesia was applied by subcutaneous injection of 6 mg/kg of lidocaine on the skull prior to skin opening. The skin was cut open over the length of a centimeter and the periosteum was removed using surgical sponges (Questalpha Sugi sponge points Ref. 31603). For both types of cranial surgery, a Microdrill (Harvard Apparatus Microdrill Model 780001 with 0.6 mm drill bits Part No 60-1000, CellPoint Scientific, Ideal microdrill burr set) was used to thin or open the skull between bregma and lambda on the right side of the sagittal suture of the skull over an area of 5 mm diameter. Overheating of the skull was prohibited by the application of cold saline every

10 sec. For skull thinning, a circular region of the skull was homogenously drilled over the entire area until the bone thickness was below 50 µm.

For acute cranial window preparations drilling was exclusively performed on the outer limit of the window and the bone plate was carefully removed under superfused conditions using a small sharp spatula (AssutSuture Spatula) leaving the dura mater intact. During the entire drilling procedures, a gentle touch of the drill to the skull was ensured. The acute cranial window preparation was superfused with 0.9% saline solution and then covered with a round coverslip (diameter 6 mm #0, Epredia CB00060RA020MNZ0), and finally sealed with dental cement[98].

### Two-photon intravital microscopy (2P-IVM) imaging of CNS

All imaging sessions were performed acutely right after surgery. 2P-IVM imaging was performed using the LaVision Biotec TriM Scope II system equipped with an Olympus BX50WI fluorescence microscope and a ×20 objective (NA 0.95; Nikon). Images were acquired at a wavelength of 920 nm from an area of 400 µm × 400 µm in x-y, scan field at a depth of 160–220 µm and z-stack of 81–111 frames with 2 µm spacing every 20 s–120s. During 2P-IVM -imaging, the acquisition of individual z-frames was synchronized with the induced mechanical ventilation of the mice, and tissue distortion and drift correction were performed using VivoFollow 2[41]. Sequences of image stacks were transformed into volume-rendered 4D images by Imaris 9.8 software. After 2P-IVM mice were euthanized with an overdose of ketamine (7.5 mg/mouse) and xylazine (0.05 mg/mouse) followed by decapitation and organ harvest.

### Tracer quantification in 2P-IVM imaging of the spinal cord

Mean fluorescence intensity of the fluorescent tracer was measured using the surface function of Imaris 9.8 software. Dura mater and subpial compartment surfaces were created based on the second-harmonic generation signal coming from the collagen type I fibers localized within the dura mater and immediately below the pia mater. Second-harmonic generation surface was later manually filtered regarding the quality and intensity of the signal and the Z position, being the dura mater surface of higher signal intensity and positioned at the top of the meninges, whereas the subpial compartment second-harmonic generation surface is found at the bottom of the meninges showing a lower signal intensity. SAS surface was manually generated between the VE-cadherin-GFP⁺ arachnoid and pia mater layers. CNS parenchyma surface was manually generated at a distance of 20–50 µm below the VE-cadherin-GFP⁺ pia mater.

### Ex vivo imaging of the meningeal layers on the surface of the brain and spinal cord

VE-cadherin-GFP knock-in mice and wild-type C57BL/6 J mice were intravenously (i.v.) injected with 30 µg of DyLight594 lectin (Vector Laboratories, Burlingame, US) to highlight the blood vessels. 15 min post lectin injection, the mice were euthanized by cervical dislocation. Under a fluorescence stereomicroscope, the brains were carefully harvested by cutting open at the midline, paying attention to not disrupting the leptomeninges on the brain. To harvest the spinal cord, the individual vertebrae of the spinal column were carefully cut lengthwise to extract the spinal cord by laminectomy, paying attention not to peel off the leptomeningeal layers. At all times the skull and vertebral column were imaged under a fluorescence stereomicroscope to check for remnants of leptomeningeal cells. The dura mater always remained on the skull and vertebral column and occasionally some leptomeningeal layers remained attached to the dura mater. The removed brains and spinal cords were placed in cold Dulbecco's phosphate-buffered saline (DPBS) on ice. Images were acquired using an Axiozoom fluorescence microscope (Carl Zeiss, Oberkochen, Germany) equipped with a Plan Neo Fluar Z ×1.0/0.25 (×7 to ×112

Zoom) objective. Images of the leptomeningeal layers covering the different regions of the CNS were acquired at a magnification of ×20 for the overviews and ×100 for the zoomed-in images. Images were processed by Imaris 9.8 software.

## Immunofluorescence staining of brain and spinal cord sections

VE-cadherin-GFP knock-in mice or C57BL/6 J wild-type mice were perfused with 2% PFA in DPBS (Merck, Darmstadt, Germany), and brains and spinal cord were removed, taking care not to destroy the leptomeninges. After collection, brains and spinal cords were fixed in 2% PFA/DPBS overnight at 4 °C. Brains were then moved to DPBS and stored at 4 °C until sectioning. Sectioning was performed using either the VT1000s vibratome (Leica, Wetzlar, Germany) for 100 µm sections, or the CryoStar NX50 (Epredia, Portsmouth, USA) for 10–20 µm thick sections. Tissues were either cut coronally (decalcified heads, decalcified vertebral columns, isolated brains, and spinal cords) or transversally (decalcified heads). 100 µm thick sections were blocked with blocking buffer (5% skimmed milk (Rapilait, Migros, Switzerland), 0.3% Triton-X-100 (Sigma-Aldrich), 0.04% NaN₃ (Fluka Chemie, Buchs, Switzerland) in Tris-buffered saline (TBS) pH 7.4) for 2 hr at RT under shaking conditions. The blocking buffer was removed, and sections were incubated with the blocking buffer containing the primary antibody overnight at 4 °C under shaking conditions. After 3× washing with DPBS, sections were incubated with blocking buffer containing the secondary antibody for 2 hr at RT under shaking conditions and protected from light. After 3 washing with DPBS, sections were placed on glass slides and allowed to dry overnight. Sections were then rehydrated with DPBS, and cover slides were mounted with embedding medium Mowiol (Sigma-Aldrich, Steinheim, Germany) prior to confocal microscopic analysis.

For some immunostainings of thin (10–20 µm) tissue sections, mice were intravenously injected with 30 µg *Lycopersicon esculentum* (tomato) lectin-DyLight 594 (Invitrogen, Cat No L32471) 20 min prior to brain and spinal cord removal allowing for vascular labeling. When using mouse antibodies, non-specific binding of antibodies to Fc receptors was prevented by incubation with 10 µg/ml of 2.4G2 (CD16/CD32) 1 hour prior to incubation with primary antibodies.

For tissue decalcification, skin and muscles were first removed from the head and vertebral column prior to postfixation in 2% PFA overnight at 4 °C. Then, tissues were rinsed with PBS and immersed in 14% EDTA (pH 7.8-8) for 7–10 days at room temperature, depending on the softness of the tissues. After the EDTA treatment, tissues were incubated in 30% sucrose for 3 days, embedded in Tissue-Tek O.C.T., and stored at –80 °C until sectioning. Cryostat sections were cut and immunostained as described for thin (10–20 µm) sections above omitting the acetone postfixation step.

Images were acquired using either LSM800 or LSM880 with Airyscan (Carl Zeiss, Oberkochen, Germany) laser scanning confocal microscopes with the ZEN software. Images were processed by Imaris 9.8 (Oxford Instruments, Oxfordshire, England) and ImageJ software (ImageJ software, National Institute of Health, Bethesda, USA).

## Antibodies for immunofluorescence staining

Primary antibodies used in this study: rabbit-anti-human/mouse E-cadherin (Cell signaling, Cat No 24E10), rat anti-mouse/human/porcine ER-TR7 (BMA biomedical, Cat No T-2109), rat anti-mouse PECAM-1 (in house, clone MEC13.3), goat-anti-human/mouse/rat/canine ALCAM (R&D systems, Cat No AF1172), rabbit-anti-mouse VE-cadherin (provided by Prof. Dietmar Vestweber, Max Planck Institute, Münster, Germany; clone VE-42), mouse anti-mouse/rat/dog/chicken α-catenin (BD, Cat No 610194), mouse anti-mouse/human/rat/dog β-catenin (BD, Cat No 610154), goat-anti-mouse/human E-cadherin (R&D systems, Cat No AF748), rabbit-anti-mouse/human claudin-11 (Novus, Cat No NBP1-82470), *Lycopersicon esculentum* (tomato) lectin-DyLight 594

(Invitrogen, Cat No L32471), mouse IgG1 κ isotype (BioLegend, Cat No 401402), rabbit IgG (R&D, Cat No AB-105-C), goat IgG (R&D, Cat No AB-108-C).

Secondary antibodies used in this study: donkey anti-rabbit IgG (H + L) Cy5 (Jackson ImmunoResearch, Cat No 711-175-152), donkey anti-rat IgG (H + L) Cy3 (Jackson ImmunoResearch, Cat No 712-165-150) and donkey anti-goat IgG (H + L) Cy3 (Jackson ImmunoResearch, Cat No 705-165-147), goat-anti-mouse (H + L) Alexa Fluor 647 (Invitrogen, Cat No A32728), donkey anti-goat (H + L) Alexa Fluor 647 (Jackson ImmunoResearch, Cat No 705-605-003).

## Transmission electron microscopy (TEM)

Three months old C57BI6/J mice were perfused through the heart with MORF fixative (2.5% glutaraldehyde, 1% paraformaldehyde in 0.1 M PIPES buffer, pH 7.4). The brain was removed and postfixed in MORF at 4 °C overnight and then coronal vibratome sections (200 µm) of the cerebral cortex were cut and stored in MORF until embedding in LR white. Vibratome sections were processed for 10 min in 0.1 M phosphate buffer, pH 7.4; 60 min in 1% osmium tetroxide in 0.1% phosphate buffer, pH 7.4; 10 min in 50% ethanol, 10 min in 70% ethanol; 10 min in 95% ethanol; 15 min in 100% ethanol; 5 min in propylene oxide; 60 min in propylene oxide:Eponate 12™ (Ted Pella); and overnight in Eponate 12™. Sections were then cut into smaller pieces and transferred to tightly sealed gelatin capsules filled with Eponate 12™ and polymerized at 60 °C for 48 h. Regions of interest were identified in 1-µm sections cut and stained with Toluidine blue. Ultra-thin sections (60 nm) were then cut on a UC7 ultramicrotome (Leica), collected in nickel mesh grids coated with a Formvar film (Ted Pella), and stained with uranyl acetate. Sections were examined in an FEI Tecnai™ G² Spirit BioTwin transmission electron microscope at 80 kV. Intercellular junctions were identified by using conventional TEM criteria[7,20,57,58]. At TJs, also designated *zonula occludens*, the space between the plasma membranes of adjacent cells was focally absent where the outer leaflets of the apposed membranes touched at points of fusion. At AJs, also designated intermediate junction, *macula adherens*, or *zonula adherens*, the space between the plasma membranes was filled with electron-dense material that extended to the cytoplasmic surface of the membranes. At gap junctions, the outer leaflets of adjacent plasma membranes were strictly parallel and separated by a uniformly narrow gap that was less than elsewhere between the cells. At all of these junctions, localized extracellular and/or cytoplasmic densities provided distinctive features that made them readily identifiable even at low magnifications.

## Statistical analysis

GraphPad Prism v9.2 software (La Jolla, CA, USA) was used to perform all statistical analysis. Precise statistical tests used are indicated in each figure legend. Data are presented as mean ± SD or ±SEM and precise $p$ values are indicated when statistical significance is reached. Figures were made with Adobe Illustrator 25.4.1.

## Reporting summary

Further information on research design is available in the Nature Portfolio Reporting Summary linked to this article.

## Data availability

All data are available in the main text or supplementary materials. Source data are provided with this paper. Imaging datasets will be made available upon request.

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

## Acknowledgements

This study was funded by the Fidelity Bermuda Foundation and the Swiss National Science Foundation (grant N° 310030_189080) to B.E. and CRSII5_213535 to S.P. and B.E. J.M. was supported by a Swiss Government Excellence Scholarship. D.M.D. was supported in part by the National Heart, Lung, and Blood Institute of the US National Institutes of Health (grants R01 HL143896, R01 HL059157, R01 HL127402). We would like to specifically acknowledge Dr. Peter Vajkoczy and Dr. Christian Uhl (Charité, Berlin, Germany) for their help and in-depth discussions in establishing and performing the acute cranial window and skull thinning preparations. We furthermore acknowledge Dr. Nicolas Page and Dr. Doron Merkler (University of Geneva, Switzerland) for providing LCMV-OVA. We finally thank Dr. Daniel Legler (Biotechnology Institute Thurgau, Switzerland) and Dr. Marcus Thelen (IRB, Bellinzona, Switzerland) for providing fluorescently labeled chemokines.

## Author contributions

Conceptualization: B.E., J.M., J.P., C.B. Methodology: J.M., J.P., M.V., E.B., P.H., P.P., S.B., J.A., D.M.D., U.D. Investigation: J.M., J.P., M.V., E.B., P.H., P.P., S.B., J.A., U.D. Visualization: J.M., J.P., D.M.D., M.V. Funding acquisition: B.E., S.P. Supervision: B.E., S.P., D.V., D.M.D., C.B. Writing of the manuscript: B.E., J.M., J.P., P.P., D.V., D.M.D.

## Competing interests

The authors declare no competing interests.
