## [Peer Review File · Nature Communications]

REVIEWERS' COMMENTS

Reviewer #1 (Remarks to the Author):

Summary of the key results: In this study, Mapunda, Paraja, Englehardt and colleagues utilize the novel observation that VE-Cadherin-GFP mouse line can be used to define two layers of leptomeningeal cells, one in the pia mater the other the arachnoid mater, in sections and 2-photon imaging as the upper and lower cellular boundaries of the subarachnoid space (SAS). The authors utilize 2P imaging with 1) different molecular size tracer injections into the cisterna magna and 2) labeled T cells in healthy adult mice to demonstrate extent of barrier properties to molecules and immune cells in the brain parenchyma and the dura. To apply use of VE-Cadherin-GFP to understand how meningeal barrier properties and SAS boundaries are altered in a neuroinflammatory disease, the authors document tracer passage and T cell trafficking, showing that the pia barrier is perturbed but the outer arachnoid barrier is not at this stage. Finally, the authors use these novel SAS landmarks with 2P in combination with the VE-CAD-GFP line to illustrate enlargement of the SAS in neuroinflammation that has not previously been documented.

Originality and significance: Recently, heightened interest in the meninges has led to numerous studies and discoveries on meninges neuroimmunology and dura lymphatics relevant to brain health, injury and disease, stemming in part from a focus on local immune cell generation and meninges immune response to CNS-derived antigens and dura located lymphatics. With the exception of a few studies, there have been comparatively few advancements in defining the functional anatomy of the mouse meninges using modern methods like 2-photon imaging as it relates to movement of molecules within and out of the SAS. The findings reported in this study are significant in that they modernize the methods that can be used to define leptomeninges borders. This is especially relevant to understanding how molecules and cells move into and out of the SAS into the brain parenchyma or the periphery, in particular the adjacent dura. There is a lack of information in this area, leading to confusion and, to a certain extent, controversy about the extent of access to the SAS and how CSF fluid and molecules move from the SAS to the periphery. In general, barriers at the arachnoid layer (which are shown in this study to be labeled by VE-Cadherin-GFP) are woefully understudied despite being documented at the cellular and functional level for decades (see Balin et al., 1986 PMID: 3782501 Rodriguez-Peralta, 1957 PMID: PMID: 13475509 and Nabeshima et al., 1976 PMID). Barrier properties at the pia, the extent to which they do or do not restrict passage of molecules or cells, has only been looked at in EM in human tissue and is largely speculative. The more information and advanced tools, like the ones identified in this manuscript, the field has, the better scientists can accurately model the meninges as an interface between the CNS and the periphery in preclinical animal models.

The need for this type of tool was underscored by a recent, high profile paper claiming the presence of '4th meningeal' layer that was met with skepticism from neuroanatomists and meninges experts, including myself. The authors, in their resubmission of this manuscript, have combined the VE-Cadherin-GFP with a putative marker of the '4th layer' Prox1-RFP and provided important characterization of the location of this cell layer within the arachnoid mater. Further, in this resubmission, they employ the VE-Cadherin-GFP to identify the time course of artifact subdural widening that occurs in preparing the skull for 2P-IV imaging. This is an important consideration for researchers studying the meninges layers, particularly as it relates to location of immune populations in healthy and disease states.

The use of the VE-Cadherin-GFP along with imaging/tracer demonstrates a previously unappreciated VE-cadherin+ cellular layer at the pia and the means to compare these two layers as barriers to tracer, as a proxy for CSF contents, in real time. This is particularly important to consider in the context of intrathecal drug delivery and compartmentalization of leptomeningeal immune cells (T cells, macrophages, dendritic cells), the latter of which is often overlooked in 2p studies of the leptomeninges and brain. Finally, application of these new methods in a neuroinflammatory disease and illustrating pial barrier breakdown (and not arachnoid barrier) as well as distension of these spaces is highly relevant to the pathophysiology of neuroinflammatory diseases like MS, where parenchymal lesions at or near the pial surface are very common human patients. Overall, this is an important and significant study that substantially advances methods and foundational knowledge of CNS barriers in the meninges and has important implications for neuroinflammatory diseases.

In this resubmission, the authors have done an outstanding, rigorous job of not only addressing all my concerns with new data (ex: AJs markers, GFAP) and quantification (ex: quantification of the location of tracer) but have added on very important new data related to Prox1+ meningeal layer, its relationship to the VE-cadherin+ arachnoid and E-Cadherin+ arachnoid barrier layer that I believe is a very important advancement to the field of meninges biology. I commend the authors on the thoroughness of the response to critiques and strongly recommend publication of this manuscript.

I only have a few very minor comments and small corrections, these do not require my re-review and can be overseen by an editor.

Minor:

Supp Fig. 1: The images and experiment are quite convincing that there artifactual enlargement of the subdural space that emerges post-surgery, important consideration for studying meninges compartments that is revealed by the experiments with the VE-Cadherin-GFP - suggest adding labels to indicate the dura blood vessel VE-Cad-GFP signal in the YZ MIP in 2nd panel from the left in A, the time course images in B and C. Also consider text labels in the YX MIP for the meningeal layers (dura and pia/arachnoid) – I recognize the author's may not want to obscure any part of the images but some

additional labels (perhaps off to the side of the image?) to guide the reader would be helpful. The labels in Fig 6C are a nice example of how this can be done.

Fig. 5D: It's unclear how the different junctions were differentiated, please provide some criteria that was applied (ex: size/length, etc) in the methods.

Typos: Ln87 – reference to Louveau, 2015 should be a numbered reference, also extra “}”.

Ln 548: add commas, should read “and its diffusion into the SAS is, as expected, blocked...”

Ln 583: typo, should be “alterations”

Signed: Julie Siegenthaler

Reviewer #2 (Remarks to the Author):

Thank you for the added experiments and careful consideration of my comments. This is an improved manuscript, and I have two additional comments.

- The new experiments to compare windows made by skull thinning vs. acute craniotomy are appreciated. The finding that there is a difference in the observation of a subdural space is important. However, the conclusion to take from this result is unclear. It is initially assumed that the thinned skull window would be less disruptive of meninges and intracranial pressure, and would capture a more realistic view of tracer flow and meningeal spaces. Yet, the authors seem to conclude that the thinned skull is creating an abnormal space due to hemorrhage or hematoma (stated in Discussion). Alternatively, the compression of the space with a coverglass may preclude the observation of a subdural space that should normally be there. Which is correct, and which is the better window for imaging this very delicate biology? Please clarify, as this conclusion will help set future imaging studies on the right foot.

- The new fluorescence intensity analyses of Figures 7 and 8 are also appreciated. However, please add to the Methods how the regions of interest were selected for analysis. In particular, how was the region of the subpial space demarcated in Imaris, given that it is such a thin layer.

Reviewer #3 (Remarks to the Author):

“VE-cadherin identifies arachnoid and pia mater cells: a missing landmark for in vivo imaging of CNS immune surveillance and neuroinflammation”

The authors have extensively revised and improved the manuscript. I particularly appreciate that in addition to the other revisions, the authors tested the observations described in the recent Science paper from earlier this year on brain meningeal layers in their VE-cadherin reporter model crossed to Prox1 reporter mice. These are important results that demonstrate some of the useful properties of the author's model.

Overall the authors have adequately addressed this reviewer's comments. Below are a couple minor points that should be added to the manuscript:

1. The authors should indicate that future studies should more thoroughly investigate the leptomeningeal barrier properties to physiological molecules such as cytokines.
2. The authors use the phrasing “non-parametric T-test” to indicate how statistics were performed in some figure captions. It would be better to describe or name the actual non-parametric test used, as the T-test is a parametric test that relies on the assumption of a normal distribution, etc. The authors should also consider adding a specific section on their statistical calculation procedures in the Methods section as well.
3. In supp. Fig. 1, the caption indicates 3kDa TRITC dextran, but the figure indicates 40kDa. Please correct.

Dear Reviewers,

Foremost we would like to thank all Reviewers for their thorough and positive feed-back, which has helped us to significantly improve the quality of our manuscript. Thank you for appreciating the significance of our findings. Please find our point-by-point reply to your final comments below. All changes and additions in our revised manuscript are highlighted in blue color.

Reviewer #1:

-Supp Fig. 1: The images and experiment are quite convincing that there artifactual enlargement of the subdural space that emerges post-surgery, important consideration for studying meninges compartments that is revealed by the experiments with the VE-Cadherin-GFP - suggest adding labels to indicate the dura blood vessel VE-Cad-GFP signal in the YZ MIP in 2nd panel from the left in A, the time course images in B and C. Also consider text labels in the YX MIP for the meningeal layers (dura and pia/arachnoid) – I recognize the author's may not want to obscure any part of the images but some additional labels (perhaps off to the side of the image?) to guide the reader would be helpful. The labels in Fig 6C are a nice example of how this can be done.

Answer: We have edited Supplementary Figure 1 accordingly by adding labels to the different meningeal layers and highlighting the dural vessels.

-Fig. 5D: It's unclear how the different junctions were differentiated, please provide some criteria that was applied (ex: size/length, etc) in the methods.

Answer: The different types of junctions were assigned based on their typical appearance in transmission electron microscopy images – this is the distance of membranes, associated electron dense material etc. We have explained these criteria in M&M and added the respective references. Due to the question of this Reviewer to elaborate on this point we have now included Dr. Donald M.

McDonald as co-author who had identified the different junctions for us in previous discussions. The included additions and references were also provided by him.

-Typos: Ln87 – reference to Louveau, 2015 should be a numbered reference, also extra “}”. *Answer: We have corrected this typo.*

-Ln 548: add commas, should read “and its diffusion into the SAS is, as expected, blocked...” *Answer: Missing commas were included.*

-Ln 583: typo, should be “alterations”
Answer: We have corrected this typo.

Signed: Julie Siegenthaler

Reviewer #2 (Remarks to the Author):

- The new experiments to compare windows made by skull thinning vs. acute craniotomy are appreciated. The finding that there is a difference in the observation of a subdural space is important. However, the conclusion to take from this result is unclear. It is initially assumed that the thinned skull window would be less disruptive of meninges and intracranial pressure, and would capture a more realistic view of tracer flow and meningeal spaces. Yet, the authors seem to conclude that the thinned skull is creating an abnormal space due to hemorrhage or hematoma (stated in Discussion). Alternatively, the compression of the space with a coverglass may preclude the observation of a subdural space that should normally be there. Which is correct, and which is the better window for imaging this very delicate biology? Please clarify, as this conclusion will help set future imaging studies on the right foot.

Answer: We thank the Reviewer for pointing out the impact of surgical procedures for brain in vivo imaging. In vivo imaging in the VE-cadherin-GFP mouse allows to see for the first time the formation of spaces between the VE-cadherin-GFP+ cells and the dura mater. We were very surprised to observe the formation of a subdural space upon skull thinning. At this time we can only point out to the potential artefacts created by acute skull thinning or cranial window preparations. We do not consider one or the other acute imaging model to be “better”. Rather further studies using this novel reporter mouse will be needed to understand if chronic window preparations are more suitable for in vivo imaging of the brain. We have elaborated on this issue in the discussion to clarify that further studies are needed to understand which surgical procedures are best to image the meningeal compartments by two-photon imaging.

- The new fluorescence intensity analyses of Figures 7 and 8 are also appreciated. However, please add

to the Methods how the regions of interest were selected for analysis. In particular, how was the region of the subpial space demarcated in Imaris, given that it is such as thin layer.

Answer: We have included a new paragraph in the Methods section detailing how the surfaces were generated for the tracer quantification in the different CNS compartments at the surface of the spinal cord.

Reviewer #3 (Remarks to the Author):

1. The authors should indicate that future studies should more thoroughly investigate the leptomeningeal barrier properties to physiological molecules such as cytokines.

Answer: We have added an additional statement in this regard in the discussion.

2. The authors use the phrasing “non-parametric T-test” to indicate how statistics were performed in some figure captions. It would be better to describe or name the actual non-parametric test used, as the T-test is a parametric test that relies on the assumption of a normal distribution, etc. The authors should also consider adding a specific section on their statistical calculation procedures in the Methods section as well.

Answer: We have edited the figure legends accordingly including the exact statistical test used in each analysis. Additionally, we have included a Statistical Analysis paragraph in the Methods section.

3. In supp. Fig. 1, the caption indicates 3kDa TRITC dextran, but the figure indicates 40kDa. Please correct.

Answer: We thank the Reviewer for pointing out this mistake. We have corrected it accordingly in Supplementary Figure 1.